# Multilayer observation and estimation of the snowpack cold content in a humid boreal coniferous forest of eastern Canada

Achut Parajuli[1,2,3], Daniel F. Nadeau[1,2], François Anctil[1,2], Marco Alves[1,2]

[1]Department of Civil and Water Engineering, Université Laval, Québec, Canada
[2]CentrEau, Quebec Water Research Centre, Université Laval, Québec, Canada
[3]Department of Environmental Science, Université du Québec à Trois Rivières, Trois-Rivières, Canada

*Correspondence to*: Achut Parajuli (achut.parajuli.1@gmail.com)

**Abstract.** Cold content ($CC$) is an internal energy state within a snowpack and is defined by the energy deficit required to
attain isothermal snowmelt temperature (0°C). Cold content for a given snowpack thus plays a critical role because it affects both the timing and the rate of snowmelt. Estimating the cold content is a labour-intensive task as it requires extracting in-situ snow temperature and density. Hence, few studies have focused on characterizing this snowpack variable. This study describes the multilayer cold content of a snowpack and its variability across four sites with contrasting canopy structures within a coniferous boreal forest in southern Québec, Canada, throughout winter 2017-18. The analysis was divided into two steps. In
the first step, the observed $CC$ data from weekly snowpits for 60% of the snow cover period were examined. During the second step, a reconstructed time series of $CC$ was produced and analyzed to highlight the high-resolution temporal variability of $CC$ for the full snow cover period. To accomplish this, the Canadian Land Surface Scheme (CLASS; featuring a single-layer snow model) was first implemented to obtain simulations of the average snow density at each of the four sites. Next, an empirical procedure was used to produce realistic density profiles, which, when combined with in situ continuous snow temperature
measurements from an automatic profiling station, provides a time series of $CC$ estimates at half-hour intervals for the entire winter. At the four sites, snow persisted on the ground for 218 days, with melt events occurring on 42 of those days. Based on snowpit observations, the largest mean $CC$ ($-2.62$ MJ m$^{-2}$) was observed at the site with the thickest snow cover. The maximum difference in mean $CC$ between the four study sites was $-0.47$ MJ m$^{-2}$, representing a site-to-site variability of 20%. Before analyzing the reconstructed $CC$ time series, a comparison with snowpit data confirmed that CLASS yielded reasonable
estimates of the snow water equivalent ($SWE$) ($R^2 = 0.64$ and percent bias (Pbias) = $-17.1$%), bulk snow density ($R^2 = 0.71$ and Pbias = 1.6%), and bulk cold content ($R^2 = 0.93$ and Pbias = $-3.3$%). A snow density profile derived by utilizing an empirical formulation also provided reasonable estimates of cold content ($R^2 = 0.42$ and Pbias = 5.17%). Thanks to these encouraging results, the reconstructed and continuous $CC$ series could be analyzed at the four sites, revealing the impact of rain-on-snow and cold air pooling episodes on the variation of $CC$. The continuous multilayer cold content time series also
provided us with information about the effect of stand structure, local topography, and meteorological conditions on cold content variability. Additionally, a weak relationship between canopy structure and $CC$ was identified.

Keywords: cold content, forest, temperature profile, snow density, snow depth

# 1 Introduction

The use of spatially distributed, process-based (physical) hydrological models has substantially improved decision-making in the area of water resources management (Wigmosta et al., 2002). The snow processes included in such models rely on the energy balance (EB) approach, since snow accumulation and melt depend on the exchanges of energy and mass between the snowpack and its surrounding environment (soil, atmosphere, and vegetation). A pioneering study on snow hydrology, led by the U.S. Army Corps of Engineers (1956), also highlighted the importance of the snowpack energy budget. Since then, the single bulk layer representation (e.g Wigmosta et al., 1994) has evolved into multilayer schemes (Gouttevin et al., 2015; Koivusalo et al., 2001; Lehning et al., 2002; Vionnet et al., 2012; Wigmosta et al., 2002). Recent studies have looked at the sources of uncertainty associated with snow models (Essery et al., 2013; Rutter et al., 2009) and revealed the importance of including some key state variables, particularly cold content, in their modelling schemes.

Cold content ($CC$) is the amount of energy required for a snow cover to reach 0°C for its entire depth. Any additional energy input translates into melting. By definition, $CC$ is a linear function of the snow water equivalent (SWE) and snowpack temperature, and is defined by:

$$CC = c_i \, \rho_s HS \, (T_s - T_m) \tag{1}$$

where $CC$ is cold content (MJ m$^{-2}$), $c_i$ is the specific heat of ice ($2.1 \times 10^{-3}$ MJ kg$^{-1}$ °C$^{-1}$), $\rho_s$ is the snow density (kg m$^{-3}$), $HS$ is the snow depth (m), $T_s$ is the snowpack temperature (°C), and $T_m$ is the melting temperature (0°C). Thus, $CC$ ranges from $-\infty$ to 0 MJ m$^{-2}$, meaning that the larger the absolute value of $CC$, the more energy required for the snowpack to eventually reach a uniform temperature of 0°C, over the entire depth of the snowpack.

$CC$ plays a central role in the timing of the snowmelt (Molotch et al., 2009), as a deep, dense, and cold snowpack requires a substantial amount of energy for snow to reach 0°C and initiate melt. As such, understanding $CC$ is essential for the accurate forecasting of water availability in demanding sectors such as agricultural systems, urban water supply (Barnett et al., 2005), and hydropower generation (Schaefli et al., 2007).

The exact determination of $CC$ requires direct observations of the snowpack temperature, density, and depth, usually collected from manual snowpit surveys. As manual collection is time-consuming, few datasets that describe snowpack $CC$ are available (e.g. Williams and Morse, 2020). For lack of a better approach, $CC$ is often estimated using one of the following three methods (Jennings et al., 2018): an empirical formulation that relies solely on air temperature (DeWalle and Rango, 2008; Seligman et al., 2014), an empirical formulation based on air temperature and precipitation (Andreadis et al., 2009; Wigmosta et al., 1994), or a residual from an energy balance model (Marks and Winstral, 2001). Jennings et al. (2018) employed snowpit data, collected at alpine and subalpine sites within the Rocky Mountains in Colorado, to study $CC$ and reported a weak relationship between $CC$ and the cumulative mean of air temperature. The authors found that newly fallen snow was responsible for 84.4% and 73.0% of the daily gains in $CC$ for alpine and subalpine snowpacks, respectively. The authors also tested the role of $CC$ in delaying snowmelt. When $CC$ was zero at 6:00 AM, the onset of snowmelt was delayed on average by 2.3 and 2.8 h at the alpine and subalpine sites, respectively. However, when $CC$ at 6:00 AM was less than 0 MJ m$^{-2}$, the onset was delayed by as

much as 5.7 h at the alpine site and 6.7 h at the sub-alpine site. This delaying effect of *CC* on melt is less marked in spring, (Seligman et al., 2014). Indeed, during spring storms, fresh snow is near 0°C, and thus adds little cold content to the existing snow cover. Under these conditions, it is therefore the addition of a new dry interstitial space (which must reach saturation) that is primarily responsible for delaying melting. Quantitatively, Seligman et al. (2014) reported that the addition of pore

spaces by dry fresh snow was responsible for 86% of the energy deficit within the snowpack of Columbia River headwaters. This suggests that even a small energy deficit has a substantial effect on the rate and timing of snowmelt. Overall, previous studies agree that the careful consideration of *CC* improves snowmelt simulations (Jost et al., 2012; Mosier et al., 2016; Valéry et al., 2014).

Little to no previous research has focused on comparing the *CC* behaviour across variable stand structures within forest

environments. Snowpack energy exchanges within a forest are different than those in open or alpine areas, as the presence of a canopy impacts snow accumulation and melt (Andreadis et al., 2009; Gouttevin et al., 2015; Mahat and Tarboton, 2012; Wigmosta et al., 2002). For instance, intercepted snow may sublimate, undergo densification, or fall beneath the canopy when maximum canopy storage is reached or when there are heavy winds present (DeWalle and Rango, 2008). Snow interception typically leads to shallower snow depths and less melt beneath the canopy (Musselman et al., 2008), even in the presence of

rain-on-snow events (Marks et al., 1998). Frequent density profiles of the snow cover allow for the tracking of unloading episodes and the identification of spatial differences of *CC* within a forest.

Despite all of the associated challenges, it is possible to simulate snow in a forested environment with some success. For instance, physically-based land surface models are regularly used to simulate snow at forested sites (e.g., Roy et al., 2013). One such example is the Canadian LAnd Surface Scheme (CLASS), which relies on a single-layer snow model based on the

energy balance. In a recent study, Alves et al.(2020) used CLASS driven by ERA5 reanalysis data to model snow depths from four dissimilar forested sites across the Canadian boreal biome. They reported average snow persistence lengths and average spring melting periods that were similar to field observations. By definition, CLASS considers the whole snowpack as a single bulk unit, and as such, is unable to simulate the multilayer behaviours that one sees in nature. One option for addressing this is to use a multilayer snow model such as SNOWPACK (Lehning et al. 2001), which was recently equipped with a detailed

canopy module (Gouttevin et al., 2015); however, even models such as this are not free of biases. For instance, Raleigh et al. (2016) tested three physically based snow models (Utah Energy Balance (UEB), Distributed Hydrology Soil Vegetation Model (DHSVM) and Snow Thermal Model SNTHERM) and reported biases in the longwave radiation estimation, ranging from −12 to +18 W m$^{-2}$. Alternatively, bulk snowpack values can be distributed between several layers. For instance, Roy et al. (2013) disaggregated CLASS-derived snow water equivalents into multilayer values at each time step, for the purpose of estimating

the specific surface area (SSA) of a snowpack. In their study, the authors reported specific surface areas ranging from 33.1 to 155.8 m$^2$ kg$^{-1}$, while attaining an acceptable root mean square error (RMSE) of 8.0 m$^2$ kg$^{-1}$ in CLASS-derived SSA for individual layers.

In view of the  lack of observational studies (particularly on *CC*) that are required to support model development in forest environments, detailed analyses of multilayer in situ snowpack *CC* are necessary. Building on Jennings et al. (2018), this study

investigates 53 snowpit-derived *CC* observations at four distinct coniferous forested sites, over the course of one winter. The temporal variability of the *CC* is also analysed by reconstructing time series that include bulk and multilayer *CC* with a 30-min time step, and combine automated snow temperature observations and bulk snow density estimates that were calculated using the CLASS model.

## 2 Methods

### 2.1 Study sites and data collection

Observations were collected in the *Bassin Expérimental du Ruisseau des Eaux-Volées* (BEREV), which is a small boreal forest catchment within Montmorency Forest, Quebec, Canada (Fig. 1). This region experiences substantial precipitation (1583 mm), with 40% falling in solid form between November and May (Isabelle et al., 2018). The boreal catchment lies in the Laurentian Mountains of the Canadian Shield and is characterized by a humid continental climate (Schilling et al., 2021). There are patches of forest clearings found within the basin due to past logging operations that have led to variability in stand structure (Parajuli et al., 2020b). Over the years, several vegetation species such as black spruce (*Picea* mariana (Mill.)) and white spruce (*Picea glauca* (Moench.)) were planted. However, the environment favoured the regrowth of balsam fir (*Abies balsamea*) stands. Isabelle et al. (2020) provide detailed information on the vegetation cover at the study site. The current analysis focuses on the four contrasting sites presented in Table 1.

Inspired by Lundquist and Lott (2010), we deployed an automated snow-profiling station at each location, composed of 18 T-type thermocouples vertically spaced 10-cm apart and of an ultrasonic depth sensor (Judd Communication, USA). These stations report the local evolution of the snowpack temperature profile and height. An additional T-type thermocouple was enclosed in a radiation shield (Fig.1c) 2 m above ground for simultaneous air temperature measurements.

**Table 1. General canopy characteristics at the four experimental sites. Sap stands for sapling, Juv for juvenile and Mat for mature.**

| Site | Mean tree height (m) | Canopy density (fraction) | LAI ($m^2\,m^{-2}$) | Forest cover |
|------|----------------------|----------------------------|----------------------|--------------|
| Sap1 | 1.8 | 0.76 | 2.8 | sapling |
| Juv1 | 8.1 | 0.76 | 3.4 | juvenile |
| Mat1 | 8.6 | 0.97 | 3.5 | mature |
| Mat2 | 12.5 | 0.71 | 2.3 | mature |

Snowpit samples at 10-cm vertical intervals (temperature profile, depth, and density profile) were collected in the vicinity of the snow-profiling stations on a weekly basis from 17 January 2018 to 24 May 2018 ($\approx$60% of the snow cover period), enabling *CC* calculation following Eq.1, and validation of the modelled output such as the snow water equivalent (*SWE*), snow density, and *CC*. Maintaining a weekly timeline was sometimes difficult due to uncontrollable circumstances such as freezing rain,

rain-on-snow events or even winter storms. During melt, from 21 April 2018 and on, it was impossible to reach all study sites because of reduced snow depths preventing the safe use of snowmobiles, except for site Juv1 that was more easily accessible from the main road. Snow-profiling stations malfunctioned occasionally (less than 1% of the time), mostly in spring. Missing values were filled with snowpit observations. An exponential moving average procedure was implemented to reduce noise in automatic snow depth observations.

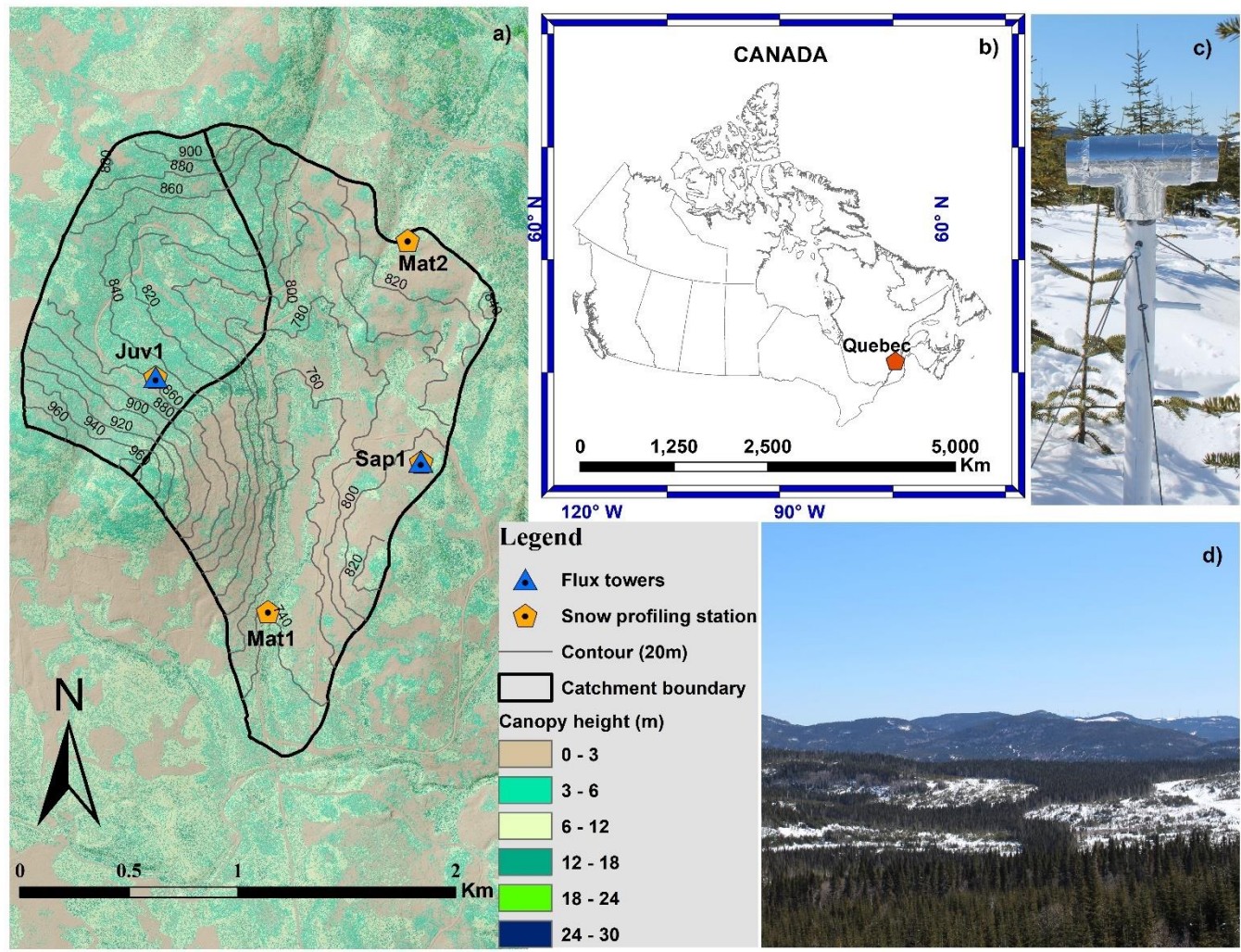


**Figure 1. Overview of the study area. (a) Basin 7 within BEREV showing the locations of the four study sites, where snowpit samples were collected and snow-profiling stations were installed, (b) the location of BEREV in eastern Canada, (c) the snow profiling station installed at site Sap1, and (d) typical winter conditions at BEREV as seen from the flux tower at site Juv1.**

### 2.2 Construction of *CC* time series

The exercise of constructing 10-cm 30-minute time series of the snowpack *CC* represents a certain challenge. On the one hand, it requires time series of the vertical profile of snow temperature, which is obtained from the snow-profiling stations. On the

other hand, time series of the snow density profile are needed as well. This is where the main difficulty lies. A simple approach would be to interpolate the density values extracted from snowpits. However, our research site has the particularity of being very snowy and experiencing many episodes delivering >10 cm of fresh snow. Thus, such interpolation would be incomplete

and error-prone given the limited number of snowpit surveys and their absence early in the season and towards the end of the melting period. Herein, it was opted to produce multilayer time series of snow density thanks to CLASS bulk simulations complemented with empirical formulations, as detailed below. Note that Figure 2 summarizes our methodological approach, describing the collection of weekly *CC* observations via snow pits (see Section 2.1) and the construction of continuous *CC* data every 30 min, which are the focus of the present section.

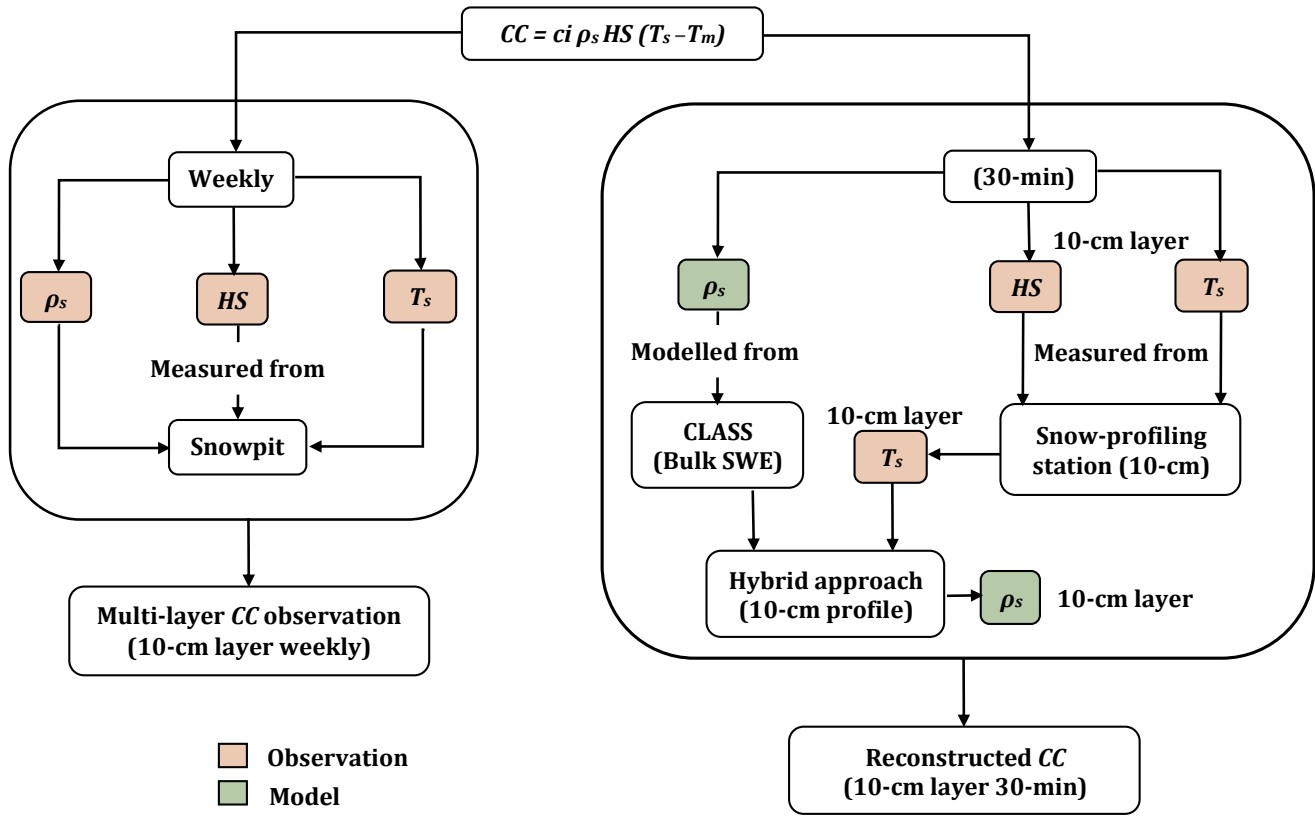

**Figure 2: Schematic representation of the methodology adopted in this study. The left column describes the acquisition of *CC* from weekly snow pits, while the right column describes the construction of continuous *CC* data every 30 min using an approach combining observations (orange cells) and modelling (green cells). $\rho_s$ stands for snow density, *HS* for snow height, and $T_s$ for snow temperature.**

**2.2.1 CLASS model**

CLASS is a physically-based land surface model that simulates the exchanges of water and energy between the Earth's surface and the atmosphere (Bartlett et al., 2006; Verseghy, 1991). It considers four distinct surface subareas: bare soil, canopy cover over bare soil, canopy with snow cover, and snow cover over bare soil (Bartlett and Verseghy, 2015; Verseghy et al., 2017).

In this analysis, CLASS version 3.6 was used in offline mode and with a 30-min time step, ensuring an uninterrupted time

series of the prognostic variables (Roy et al., 2013) hereby allowing  the inclusion of multiple soil layers and accounts for snow interception, snow thermal conductivity, and snow albedo, as described in Bartlett et al. (2006). The following subsections describe the meteorological forcing data required to run CLASS, and the methodology (CLASS + snow-profiling station) to produce single- and multi-layer time series of snow density, following Andreadis et al. (2009).

### 2.2.2 CLASS setup and forcing

The meteorological inputs required to run CLASS include precipitation rate, wind speed, air specific humidity, incoming shortwave and longwave radiation, air temperature, and surface atmospheric pressure (Table 2). As CLASS is designed to explicitly consider the energy exchanges between the soil surface, vegetation, snowpack, and atmosphere, above-canopy meteorological forcings are used. The model accounts for local effects associated with the presence of a canopy (e.g., attenuation of incident radiation, etc.), and incorporates user-defined parameters such as tree height, canopy density, and leaf

area index (Table 1).

**Table 2. Local availability of meteorological forcing data for use in CLASS simulations. Precipitation data are from Environment and Climate Change Canada (ECCC), weather station 7042388.**

| Inputs | Sites | | | | |
|---|---|---|---|---|---|
|  | Sap1 | Juv1 | Mat1 | Mat2 | ECCC |
| **Meteorological inputs** | | | | | |
| Precipitation rate |  |  |  |  | × |
| Incoming shortwave radiation | × | × |  |  |  |
| Incoming longwave radiation | × | × |  |  |  |
| Air temperature | × | × | × | × |  |
| Surface atmospheric pressure | × | × |  |  |  |
| Wind speed | × | × |  |  |  |
| Air specific humidity | × | × |  |  |  |
| **Vegetation information** | | | | | |
| Leaf area index (LAI) | × | × | × | × |  |
| Canopy height | × | × | × | × |  |
| Canopy density | × | × | × | × |  |

These inputs serve to derive the energy budget components of the soil, vegetation, snowpack, and atmosphere. After solving

the energy balance equation for the abovementioned interfaces, the residual term (the available energy for melting or refreezing ($Q_m$, in W m$^{-2}$)) is derived. The amount of available melting/freezing energy next serves to compute the meltwater mass $M$ given as:

$$M = Q_m \left( \rho_w L_f B \right)^{-1} \tag{2}$$

where $M$ is the melt rate (m s$^{-1}$), $\rho_w$ is the density of water (kg m$^{-3}$), $L_f$ is the latent heat of fusion (J kg$^{-1}$), and $B$ is the thermal quality of the snowpack, which is defined as the energy required to melt a unit mass of snow divided by energy required by unit mass of ice at 0°C, and is dimensionless. The melting or refreezing of the snowpack is associated with the available energy ($Q_m$). A positive value of $Q_m$ might result in the melting of the snowpack, given that the available energy is large enough to eliminate the cold content and induce melt. However, a negative value of $Q_m$ contributes to the refreezing of liquid water or simply adds to the *CC*.

Precipitation rates were determined using a GEONOR weighting gauge equipped with a single Alter shield approximately 4 km north of the study area, and were considered to be uniformly distributed throughout the catchment, which is a reasonable but imperfect assumption. Given the known wind-induced bias associated with this type of gauge (Pierre et al., 2019), a simple adjustment was applied. This adjustment involved twice-daily manual precipitation observations from a Double Fence Intercomparison Reference (DFIR) setup close by, as in Parajuli et al. (2020a). Although topography affects precipitation, no correction for height differences between stations was applied, as these are relatively small (< 200 m). Vegetation parameters were extracted at each site from a LiDAR dataset. Wind speed, air specific humidity, shortwave and longwave radiation, and surface atmospheric pressure measurements were taken from flux towers at sites Sap1 and Juv1. Comparable data were unavailable at sites Mat1 and Mat2 (Table 2).

This study was carried out in a small experimental watershed with an area of 3.49 km$^2$, where the sampling sites were close to one another (Fig.1) but had distinct characteristics (Table 1). Given the similarity (more or less) in canopy structure, we opted to use the inputs recorded at site Juv1 to run CLASS simulations at sites that lacked direct measurements of meteorological variables (Table 2). Here we assumed negligible differences in the above-canopy inputs between sites Juv1, Mat1 and Mat2. The following sub-section highlights the steps adopted to generate the multilayer density estimates needed to calculate the *CC* time series for all snow layers.

### 2.2.3 Reconstruction of multilayer snow density time series (a hybrid approach)

The empirical formulation described in Andreadis et al. (2009), based on Anderson (1976), is used to reconstruct multiple layer snow density estimates by combining the CLASS-derived *SWE* estimates (hereafter referred to as the hybrid procedure). Fresh snow density follows the formulation from Brun et al. (1989), who developed the method using data collected in the French Alps. We initialized the density of fresh snow by imposing a minimum snow density of 76 kg m$^{-3}$, based on available snowpit observations, and then using the equation:

$$\rho_f = \max[(109 + 6(T_a - 273.15) + 26\sqrt{u_m}), 76] \tag{3}$$

where $\rho_f$ is the density of fresh snow (kg m$^{-3}$), $u_m$ is the wind speed (m s$^{-1}$), and $T_a$ is the air temperature (K). As snow undergoes compaction due to metamorphism and the increasing weight of overlying snow, density is assumed to increase according to the following rate:

$\quad \frac{\Delta \rho_s}{\Delta t} = (CR_m + CR_o)\rho_s$ (4)

where $t$ is time (s), $CR_m$ is the snow compaction due to metamorphism (kg m$^{-3}$ s$^{-1}$), and $CR_o$ is the compaction due to the weight of overlying snow (kg m$^{-3}$ s$^{-1}$). $CR_m$ is then calculated as (Andreadis et al., 2009):

$$CR_m = 2.778 \times 10^{-6} c_3 c_4 e^{-0.04 \times (273.15 - T_s)}$$

$$\begin{cases} c_3 = c_4 = 1 & \rho_s \leq 150 \text{ kg m}^{-3} \\ c_3 = e^{-0.046(\rho_s - 150)} & if \; \rho_s > 150 \text{ kg m}^{-3} \\ c_4 = 2 & \rho_s > 150 \text{ kg m}^{-3} \end{cases}$$ (5)

and $CR_o$ is calculated as (Andreadis *et al.* 2009):

$$CR_o = \frac{P_s}{n_0} \times e^{-c_5 (273.15 - T_s)} e^{-c_6 \rho_s}$$ (6)

where $n_0 = 3.6 \times 10^{-6}$ N s m$^{-2}$ is the snow viscosity, $c_5 = 0.08$ K$^{-1}$, $c_6 = 0.021$ m$^3$ kg$^{-1}$ and $P_s$ (N s m$^{-2}$) is the load pressure for each layer. The load pressure is defined as:

$$P_s = \frac{1}{2} g \rho_w (SWE_{ns} + f SWE_s )$$ (7)

where $g$ is the acceleration due to gravity 9.8 m s$^{-2}$, $\rho_w$ is the density of water (kg m$^{-3}$), $SWE_{ns}$ and $SWE_s$ are the amount of new snow and the snow (derived from CLASS) within the snowpack layer (mm w.e.), respectively, and $f$ is the empirical compaction coefficient taken as 0.6 (Andreadis et al., 2009).

## 3 Results

### 3.1 Local meteorological conditions

Figure 3 displays daily air temperature and wind speed observations. The shaded zone and site-specific dots illustrate the temporal distribution of the manual snow surveys. Air temperature measurements were taken at 2 m above ground. Wind speed sensors were located 3 m and 2 m above ground at sites Sap1 and Juv1, respectively. To compensate for this height difference and enable fair comparisons between sites Sap1 and Juv1, wind speed measurements at site A1 were adjusted to a 2-m height, assuming a log profile. As expected, the sapling site (mean canopy of 1.8 m) experienced higher wind speeds (mean 1.3 m s$^{-1}$) than the juvenile one (mean canopy of 8.1 m and mean wind speed of 0.12 m s$^{-1}$). Air temperatures were homogenous from site to site, with average values of −6.1 °C, −6.3 °C, −6.9 °C, and −6.5 °C at sites Sap1, Juv1, Mat1, and Mat2, respectively.

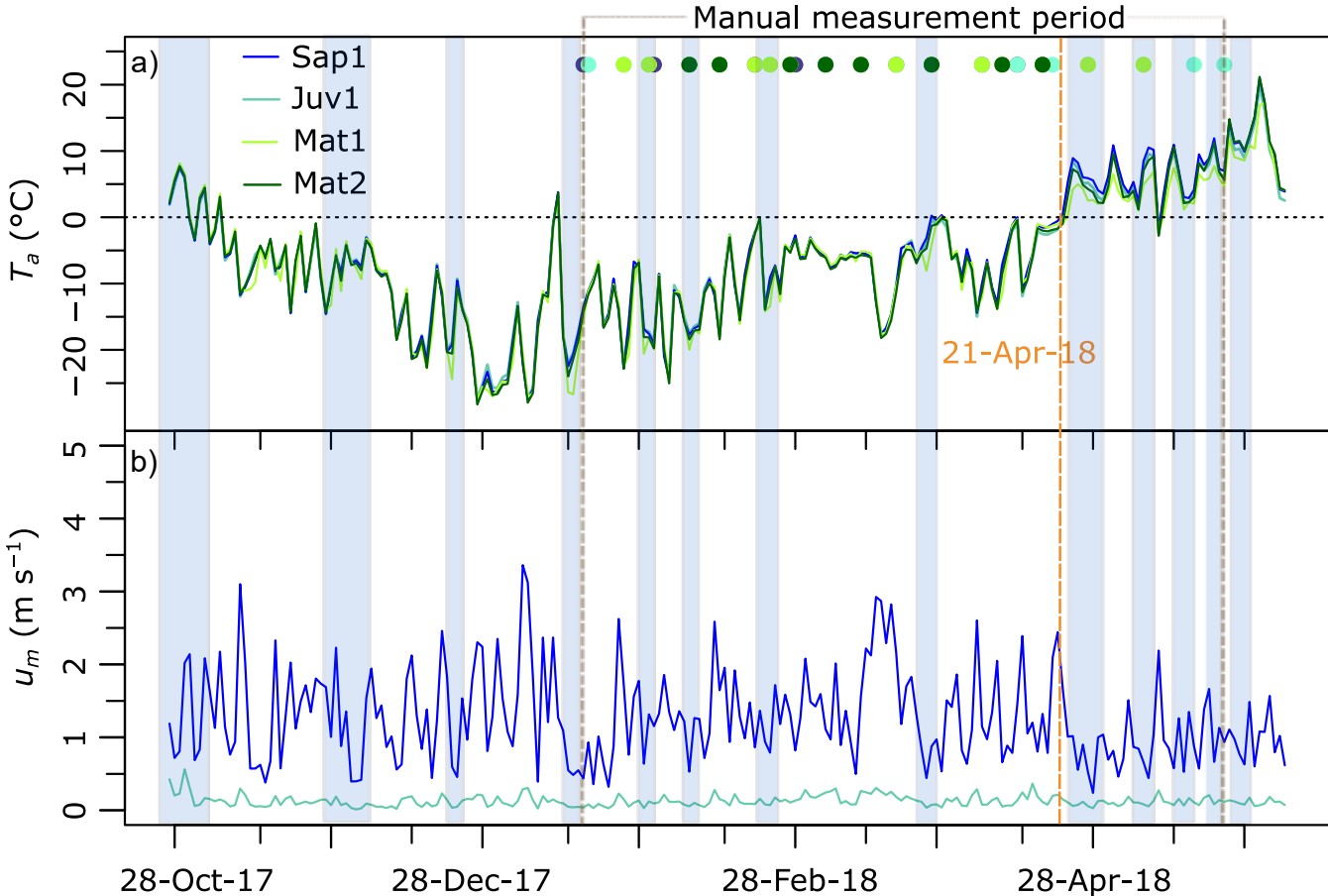

**Figure 3. (a) Daily 2-m air temperature ($T_a$) observations for all the study sites. (b) Daily 2-m wind speed ($u_m$) for sites Sa1 (sapling) and Juv1 (juvenile). Shaded areas denote periods of low wind speeds (< 0.8 m s⁻¹). Coloured points illustrate site specific snowpit surveys. Spring melt started on 21 April 2018.**

### 3.2 *CC* observations from snowpit surveys

Figure 4 illustrates 10-cm *CC* derived from the snowpit surveys. One has to sum up all the values of a single profile to find the total *CC* for a specific date. Variability in snow depth, mainly induced by contrasting canopy structure, is indicated in Figure 3. When comparing layer-wise (10 cm) differences across our sites (Sap1, Juv1, Mat1, and Mat2), the lowest (−0.013 MJ m⁻²) and peak (−0.67 MJ m⁻²) *CC* both occurred at site Juv1. Unlike for spring melt, when *CC* is low and relatively uniform, the accumulation period portrays substantial layer-wise variability structured around three distinct layers. For instance, sites Sap1, Juv1, Mat1, and Mat2 reported 9, 15, 12, and 5 observations of *CC* that were below −0.35 MJ m⁻². Note that the occurrence of large magnitude of *CC* values was not always confined to the topmost layer, as the layer just beneath the top layer also exhibited such magnitude (see Fig. 4, week 10, site Mat1 at a snow depth of 106 cm, for example). However, the layers that are close to the ground experienced smaller magnitude of *CC* throughout the winter. Excpet for site Juv1, peak *CC* occurred in early February (Fig. 5 and Table 3), when the minimum daily air temperature fell to about –25°C (Fig. 3). At

that time, the magnitude of *CC* was highest at site Sap1, which also fostered the deepest snowpack (128 cm). Total *CC* time series highlight the variability of *CC* across the four study sites (Fig. 5). Site Juv1 attained its peak *CC* on mid March (Table 3).

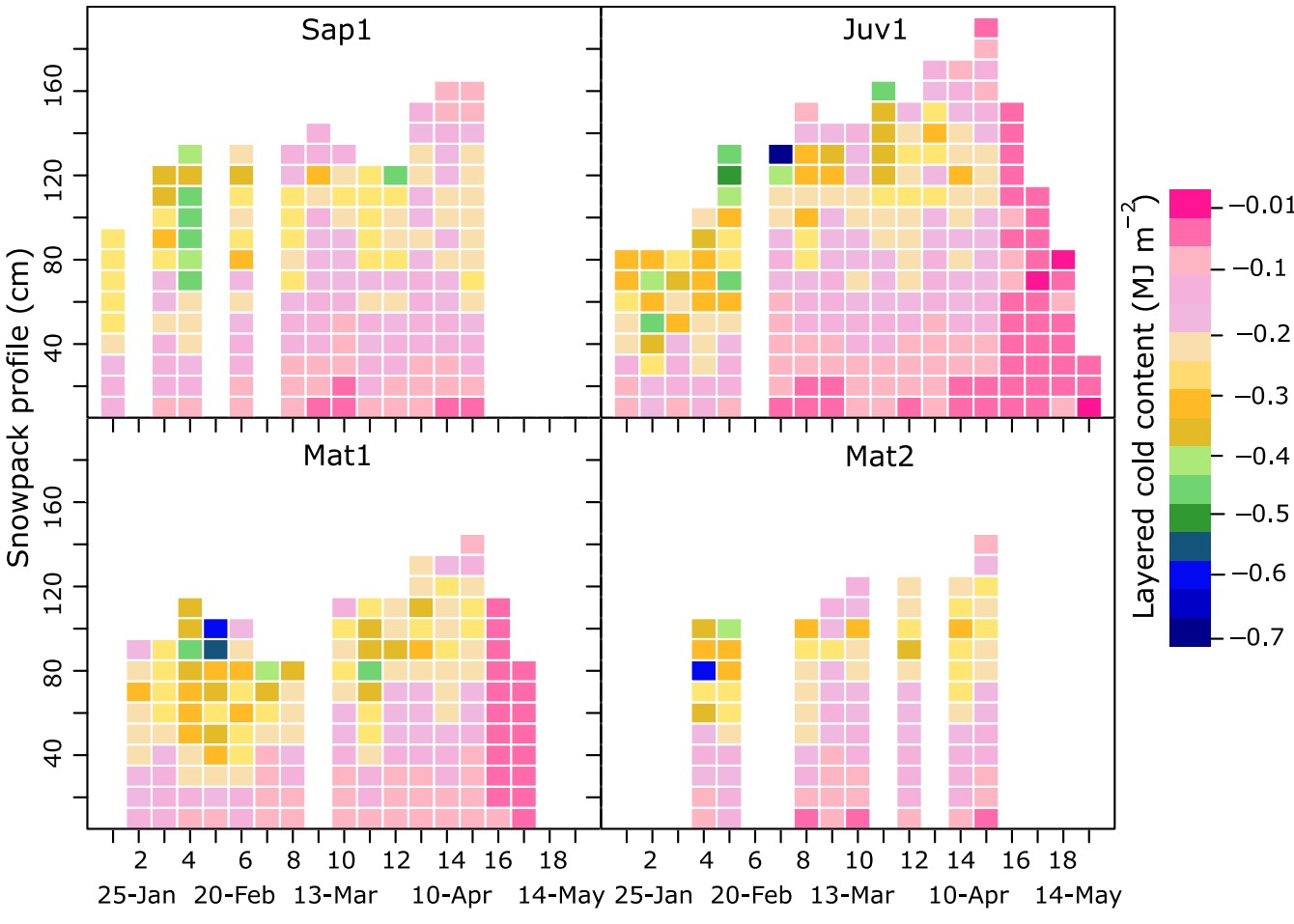

**Figure 4. Weekly 10-cm *CC* observations from snowpit surveys. The light blue shading indicates active spring melt. The colour bar indicates *CC* values in MJ m$^{-2}$**

Overall, maximum snow depth occurred at site Juv1 (194 cm), which also experienced the largest magnitude of mean total *CC* ($-2.62$ MJ m$^{-2}$), as a thicker snowpack can hold more *CC* (Table 3). For its part, Mat2 experienced the smallest maximum snow depth (142 cm) and the lowest magnitude of mean total *CC* ($-2.15$ MJ m$^{-2}$). The *CC* difference across sites reached 0.47 MJ m$^{-2}$ in total cold content, representing a variability of 20%.

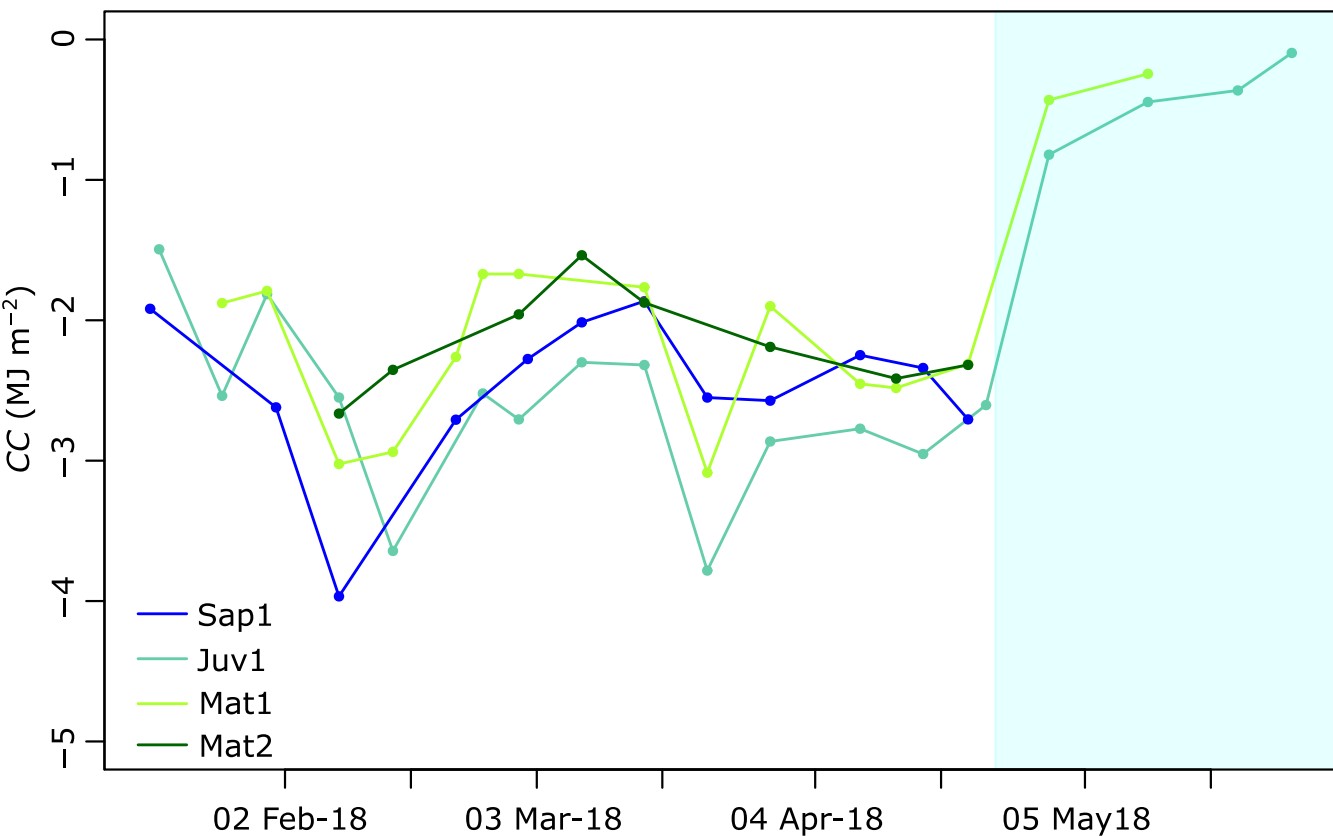

**Figure 5. Weekly snowpack total *CC* from snowpit surveys. The light blue shading represents active spring melt.**

255

**Table 3. Peak total *CC*, date of occurrence, snow depth, mean tree height at the site, maximum snow depth, and mean total *CC* over a period of 15 weeks.**

| Sites | Peak total $CC$ (MJ m$^{-2}$) | Date of occurrence of peak total $CC$ | Snow depth at peak total $CC$ (cm) | Tree height (m) | Maximum snow depth (cm) | Mean total $CC$ (MJ m$^{-2}$) |
|---|---|---|---|---|---|---|
| Sap1 | −4.05 | 2018-02-07 | 128 | 1.8 | 163 | −2.45 |
| Juv1 | −3.77 | 2018-03-20 | 173 | 8.1 | 194 | −2.62 |
| Mat1 | −3.24 | 2018-02-13 | 95 | 8.6 | 143 | −2.26 |
| Mat2 | −2.66 | 2018-02-07 | 100 | 12.5 | 142 | −2.15 |

### 3.3 Analysis of reconstructed *CC* time series

### 3.3.1 Snow density modelling

A comparison of CLASS snow simulations and snowpit (manual) observations reveals that CLASS is reasonably able to simulate bulk snow density (Fig. 6; $R^2 = 0.71$, Pbias = 1.6%), *SWE* ($R^2 = 0.64$, Pbias = −17.1%), and *CC* ($R^2 = 0.93$, Pbias = −3.3%). However, when comparing CLASS outputs, it appears that *SWE* is more underestimated than the snow density and *CC* (Fig. 6).

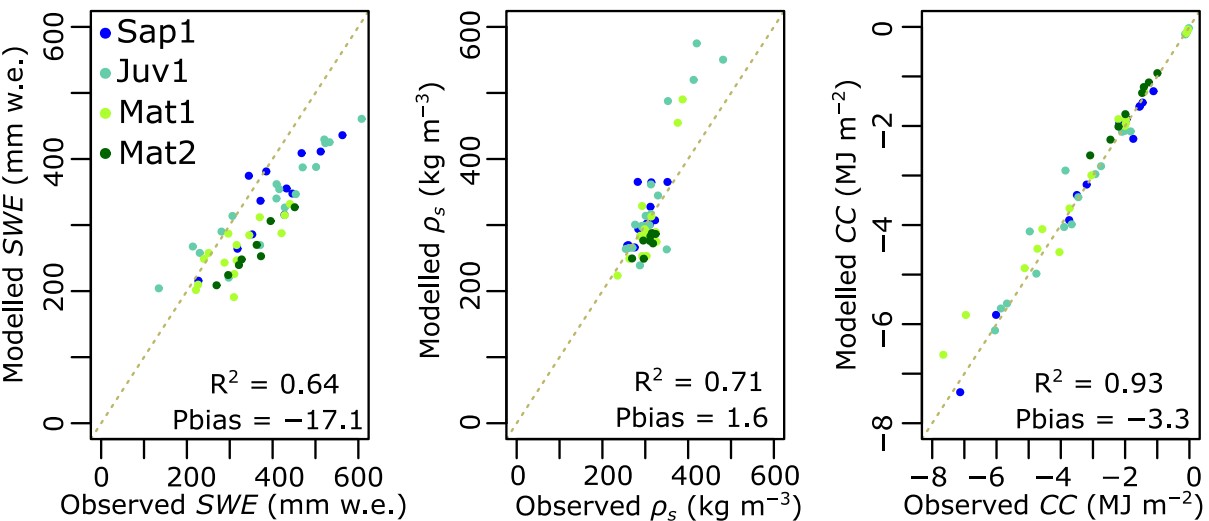

**Figure 6. Observed versus CLASS-simulated bulk values of SWE, snow density, and *CC*. $R^2$ denotes the coefficient of determination and Pbias (%) represents the percent bias.**

Nevertheless, after confirming that CLASS was efficient in modelling bulk snow cover variables, we moved forward with the next step in our methodolgy. We adopted a "hybrid procedure" in which we reproduced a vertical structure following Andreadis et al. (2009). We compared layer-by-layer, simulated and observed *CC* and snow density (Fig. 7) and determined that on average (when all sites are considered), the empirical density and observed snow temperature yielded reasonable *CC* estimates ($R^2 = 0.54$ and Pbias = −21.1 %). At each site, reasonable *CC* values were achieved over the entire profile ($R^2 = 0.67, 0.45, 0.58$, and $0.29$ and Pbias = −15.8 %, −19.7 %, −23.5 %, and −31.6 % at sites Sap1, Juv1, Mat1, and Mat2, respectively) (Fig. 8).

As displayed in Figure 7, we structured the 10-cm layers into a 3-layer scheme formed of the top (upper 40 cm), bottom (lower 30 cm) and middle layers (remainder). *CC*s for the top layer were more difficult to simulate ($R^2 = 0.57, 0.35, 0.51$, and $0.23$) than for the other two layers. Bottom-layer simulations were the most successful ($R^2 = 0.98, 0.91, 0.91$, and $0.89$). Density simulations over the entire profile behaved similarly ($R^2 = 0.5. 0.46. 0.51$, and $0.34$ and Pbias = −7.7 %, −9.2 %, −12.2 %, and −16.9 % at sites Sap1, Juv1, Mat1, and Mat2, respectively). The performance of density simulations, for each layer examined individually, was much less successful, mostly for the top layer ($R^2 = 0.06, 0.30, 0.34$, and $0.02$ and Pbias = −35.9 %, −34.5 %, −34.5 %, and −48.2 % for sites Sap1, Juv1, Mat1, and Mat2, respectively). However, snow density estimates had a low

impact on *CC* performance, as described previously (Fig. 6). These results were deemed sufficient for moving forward with the fine-scale temporal analysis of the reconstructed *CC*.

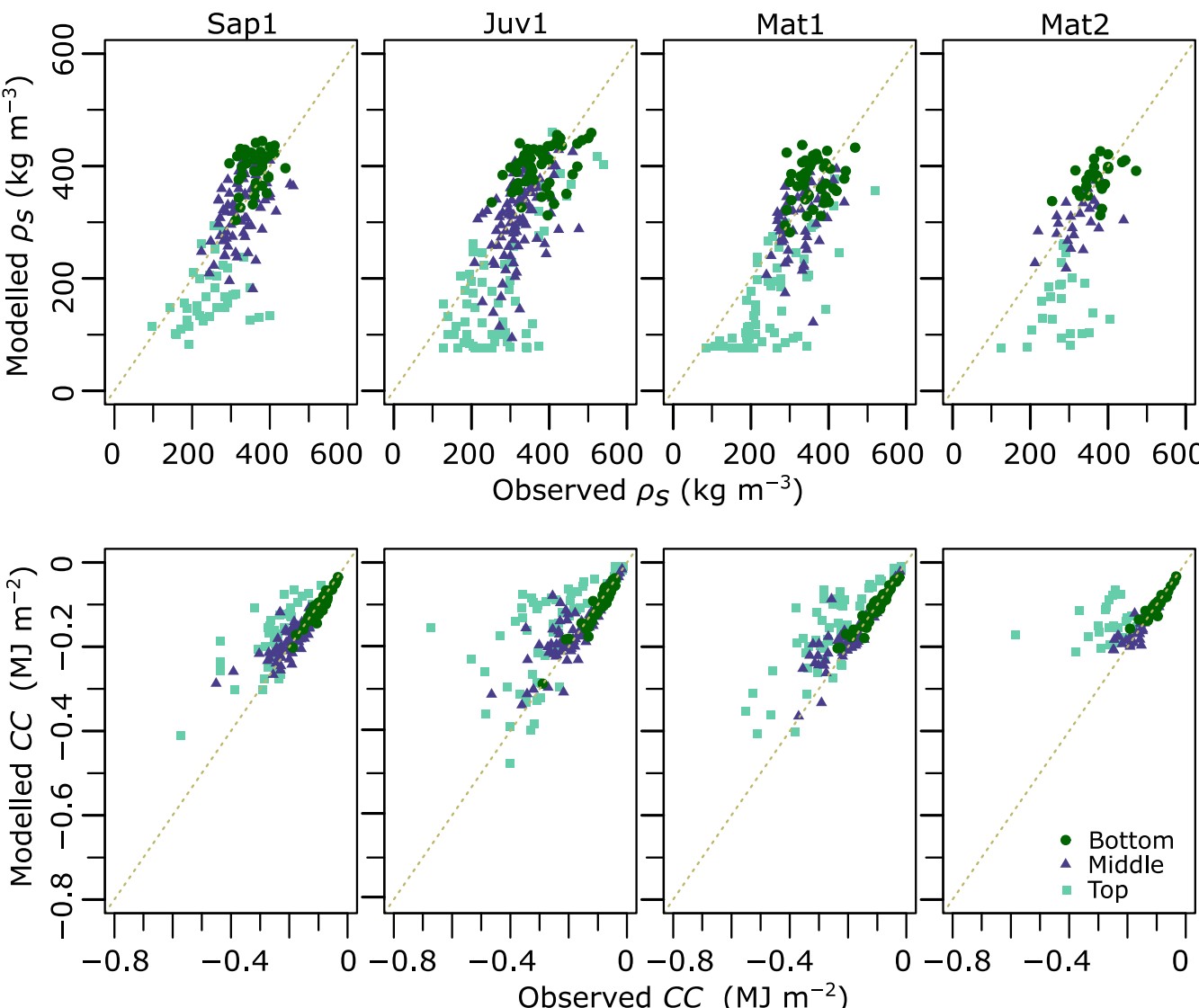

Figure 7. Observed versus modelled snow density and CC derived from the empirical formulation described in subsection 2.2.3 across the four sites. Snowpack layers are aggregated into three classes: top (upper 40 cm), bottom (lower 30 cm), and middle (remainder).

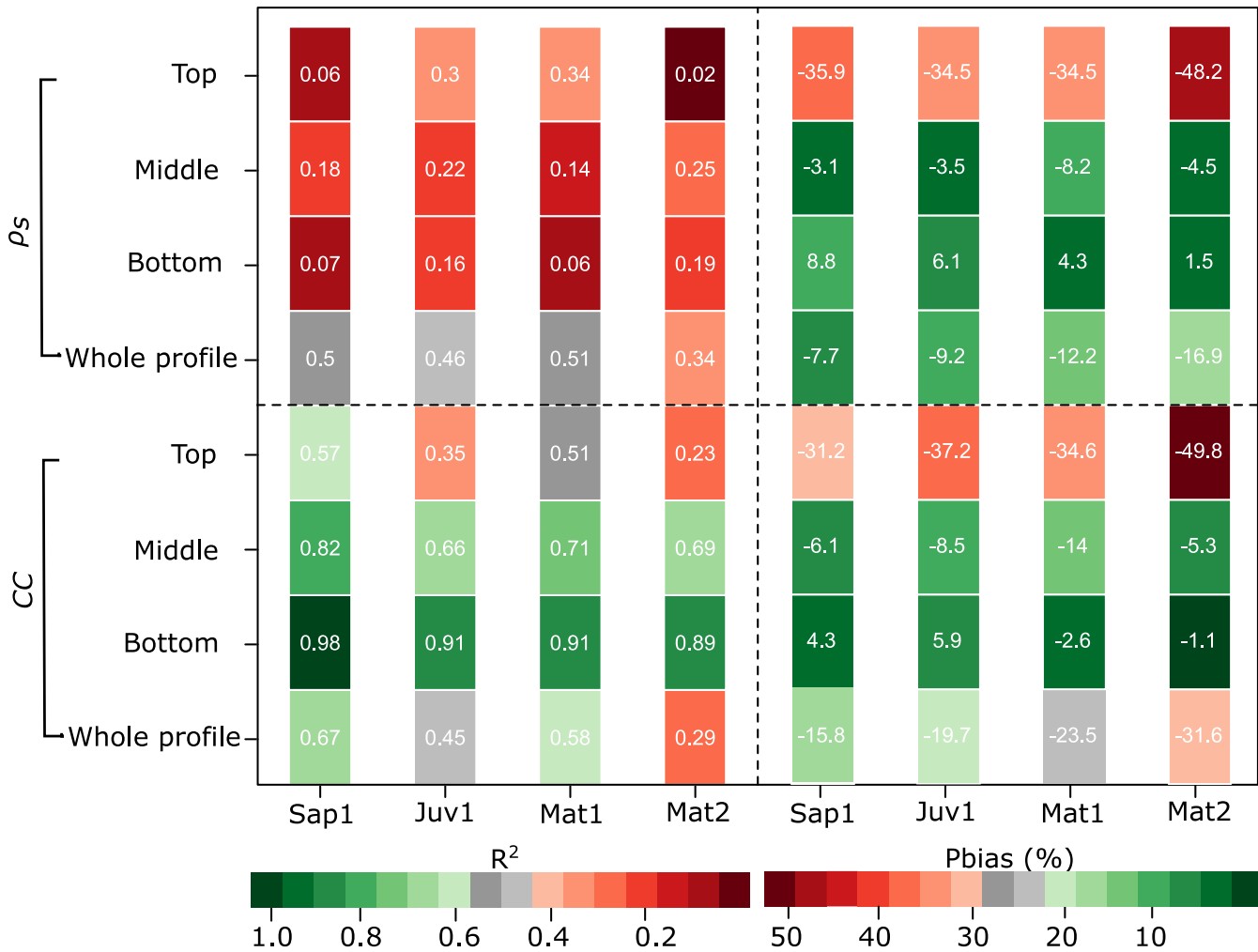

**Figure 8. Performance of *CC* and density simulations following the adopted hybrid procedure, coefficient of determination ($R^2$, left) and percent bias (Pbias, right). Pbias colour bar applies to both positive and negative values.**

### 3.3.2 Reconstructed CC time series

Figure 9 illustrates reconstructed multilayer 30-min *CC* time series. Although larger peaks (−6.9 MJ m$^{-2}$ for site Sap1) and smaller average values (−1.8 MJ m$^{-2}$ for site Juv1) are observed, the high-resolution *CC* time series that were derived using the hybrid procedure follows a pattern similar to the *CC* observations presented in section 3.3.1 (Fig. 9, Fig. 10a, and Table 2). Additionally, sites with less vegetation (site Sap1) experienced higher peak *CC* than sites with mature forest (Mat2) (Fig. 9). Notably, the rain-on-snow episodes that occurred on 11 January, 20 February, and 30 March 2018 (thin vertical bands of low *CC*) were absent from the weekly series shown in Figure 4. Sites Sap1, Mat1 and Mat2 had a shallower snowpack and the rainfall penetrated deeper, resulting in a reduced *CC* throughout the snowpack. Contrarily, at site Juv1 which had a deeper snowpack, similar rain penetration into the snowpack was only observed on 11 January 2018. All snowpacks became

300 isothermal from 21 April 2018 onwards, indicating the onset of spring melt. Some cold spells during spring melt were also noticeable, especially for the shallower snowpacks (Sap1, Mat1 and Mat2).

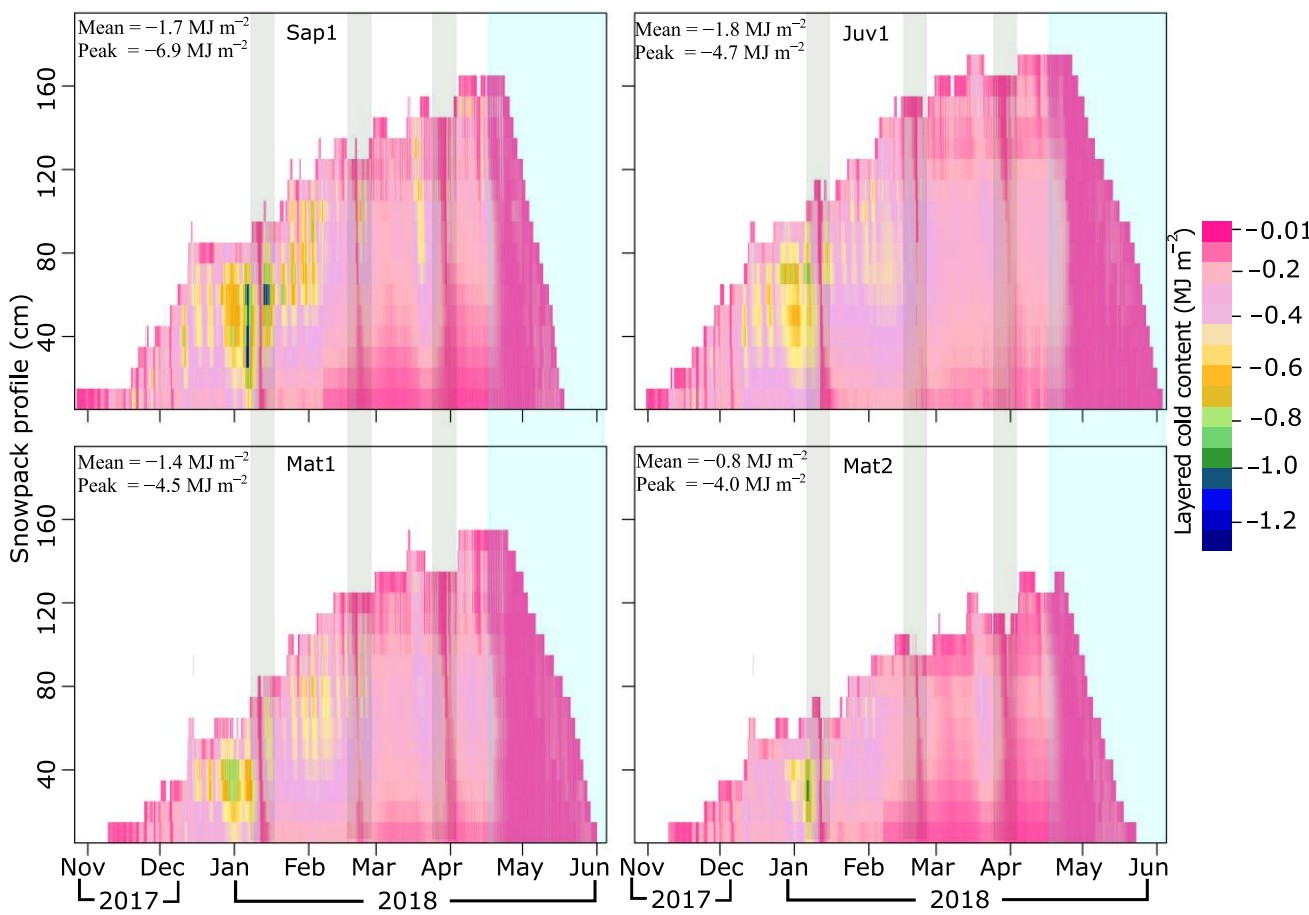

**Figure 9. Seasonal variability of 10-cm *CC* simulations stored at 30-min time intervals. The colour bar indicates *CC* values in MJ m⁻². Light green shading represents rain-on-snow events and light blue shading represents melt.**

305 Due to differences in snow accumulation and melt patterns (Fig. 9), mostly induced by differences in vegetation (Table 1), there is noticeable site-to-site variability in *CC* (Fig. 8). The detailed variability of total *CC* across the four forested sites is presented in Figure 10, along with snow depth. The magnitude of total *CC* at site Juv1 was larger than at Sap1 approximately 60% of the time. At site Mat1, this fraction drops to 32%.

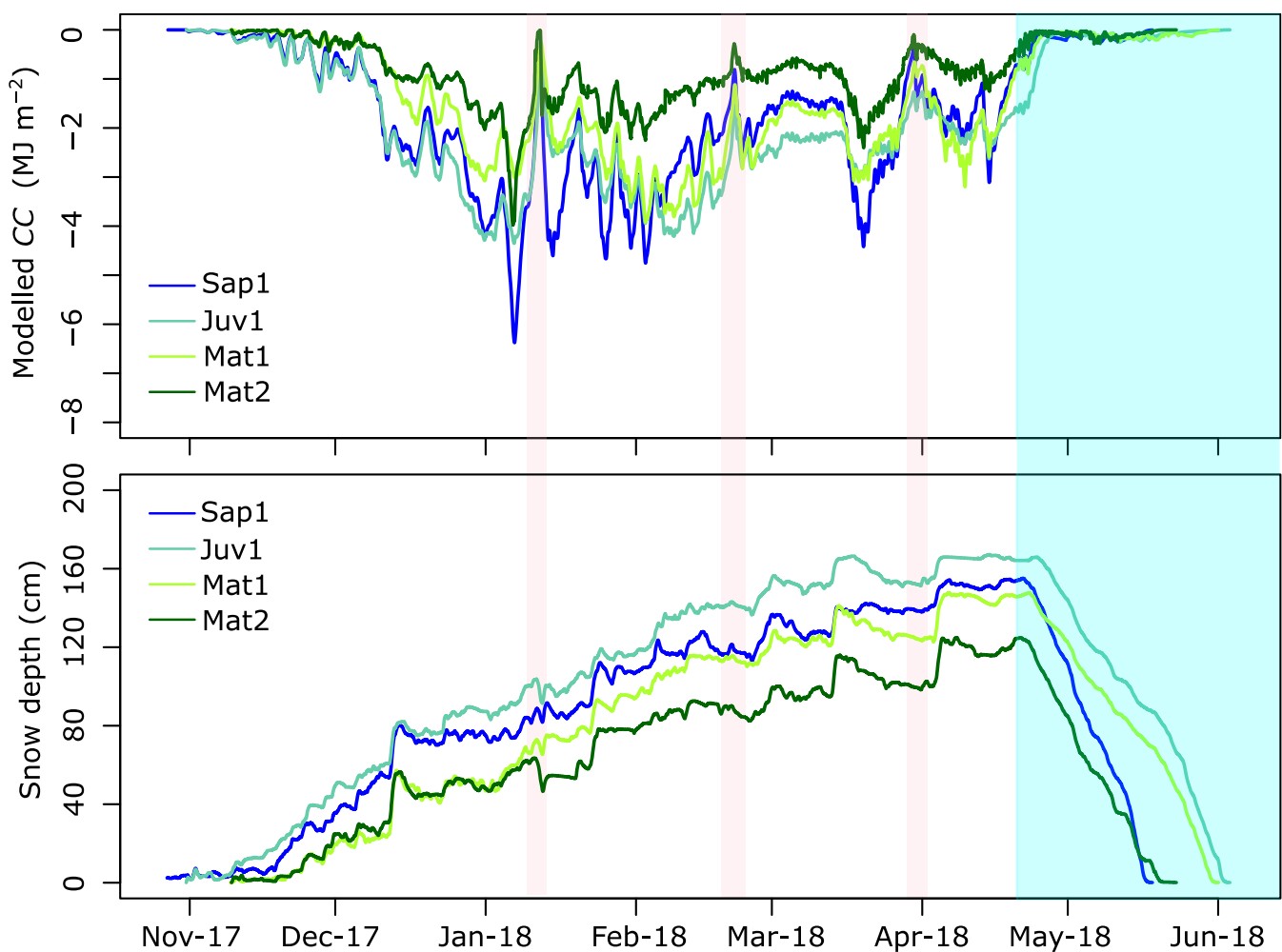

310 **Figure 10. (a) Total estimated *CC* from the reconstructed time series. An exponential moving average was applied for noise removal. (b) Snow depth observations. Spring melt period and rain-on-snow events are highlighted with light blue and light pink color, respectively.**

To shed light on site-to-site *CC* variability, Figure 11 presents the average available energy $Q_m$ (W m$^{-2}$) for melting or refreezing derived from CLASS. To facilitate the comparison, the figure divides the cold season into the winter accumulation

315   (WA) period (excluding rain-on-snow), the spring melting (SM) period, rain-on-snow (RS) events, and cold air pooling events, with low wind speed and nighttime air temperatures at site Mat1 smaller than at the other sites during the accumulation period (CP) (see also Fig. 3). During SM and RS, considerable amounts of energy were available, that might have depleted *CC* and initiated snowmelt. Conversely, WA and CP might have contributed to the refreezing of liquid water or to an increase in *CC* (Fig. 11). As expected, more melting energy was available at site Sap1 (lower vegetation) during SP (43.0 W m$^{-2}$) and RS

320   (46.0 W m$^{-2}$). During WA, the mean of available melting/freezing energy was more or less similar at all sites and ranged from −6.6 (Sap1) to −7.2 (Juv1) W m$^{-2}$. Finally, during CP, the mean melting/refreezing energy was the smallest at Mat1 (−6.6 W m$^{-2}$) and the largest at Juv1 (0.4 W m$^{-2}$), the site with the highest elevation (849 m ASL).

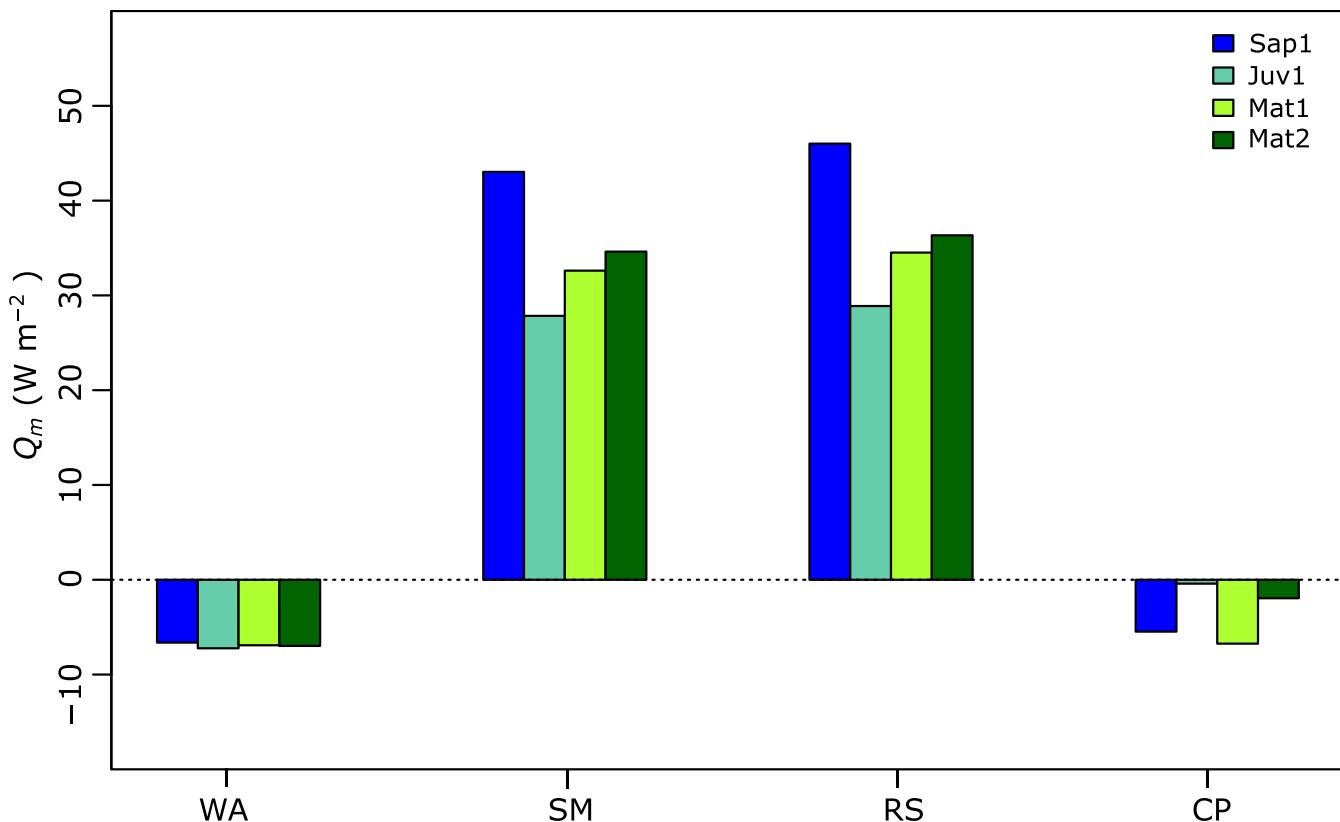

**Figure 11: Available melting/freezing energy at our sites. WA denotes the winter accumulation period, SM the spring melting period, RS the rain-on-snow events and CP events with low wind speed coinciding with smaller air temperature at site Mat1.**

Based on the Pearson's correlation coefficient ($r$), we examined the relationship between $CC$ and the snow density ($\rho_s$), snow depth ($HS$), snowpack temperature ($T_s$), and air temperature ($T_a$; Fig. 10).. Snowpack temperature ($r = 0.83$ and $0.69$) and air temperature ($r = 0.56$, and $0.66$) exhibited a positive correlation. Conversely, snow depth ($r = −0.5$ and $−0.45$) exhibited a negative correlation, whereas snow density ($r = 0.4$ and $0.24$) showed a weak relationship.

Next, we examined the relationship between each of the above-mentioned variables and $CC$ at the individual sites. This was done to identify any trends in the site-wise relationship between $CC$ and $\rho_s$, $HS$, $T_s$, and $T_a$. A decreasing trend in the correlation coefficient ($r$) with increasing mean tree heights was observed when we examined the snow temperature and the reconstructed cold content for each site ($r = 0.75$, $0.69$. $0.67$ and $0.60$ for sites Sap1, Juv1, Mat1, and Mat2, respectively). Beyond that relationship, we did not identify any site-wise trends between $CC$ and the other variables, thereby suggesting a weak dependency on forest structure in the relationship between $CC$ and other pertinent variables.

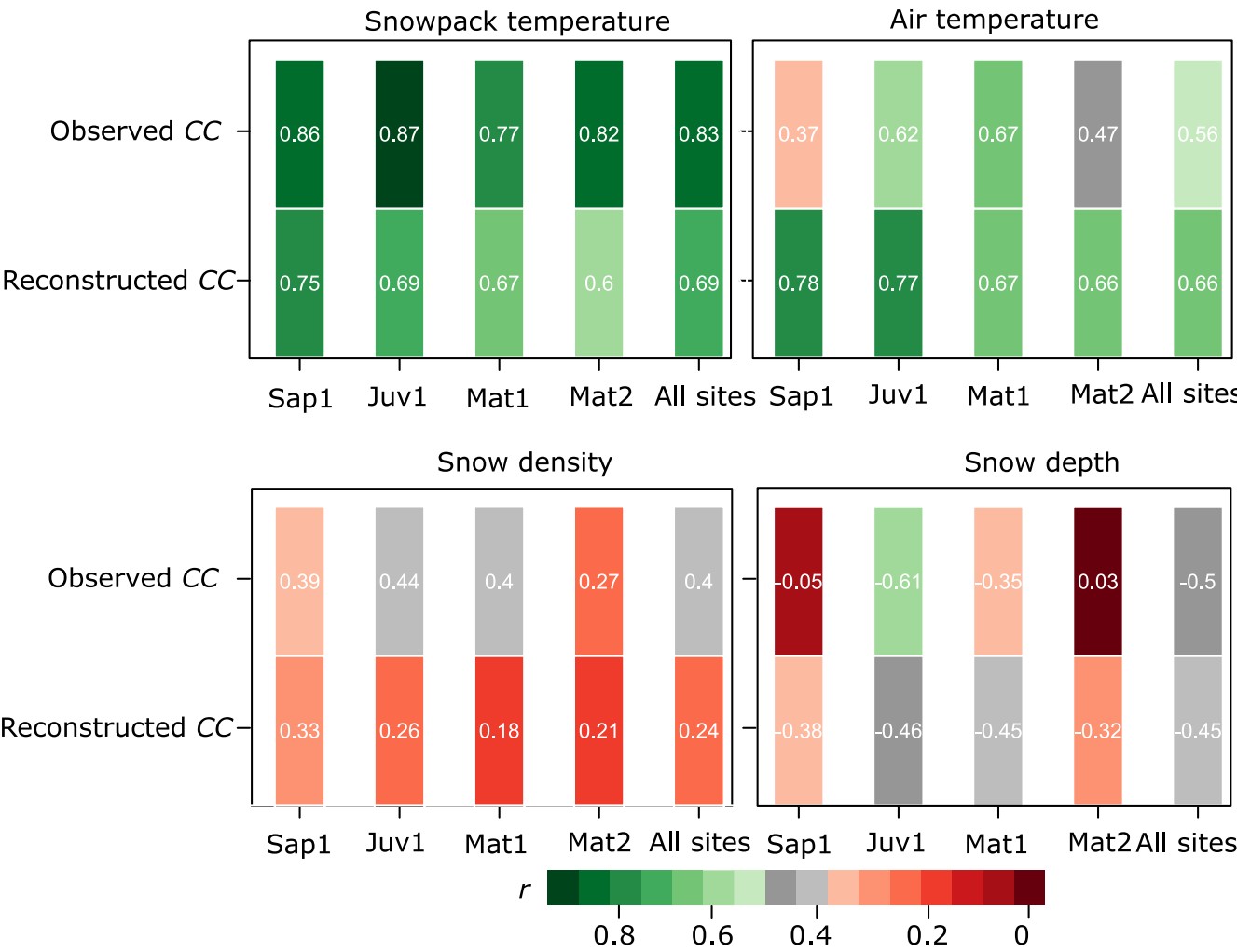

**Figure 12. Inter-relationship between *CC* and snow density, snow depth, snowpack temperature and air temperature. The colour bar represents the absolute value of the Pearson's correlation coefficient (*r*).**

## 4 Discussion

### 4.1 *CC* observations

As illustrated in Figures 3 and 4, the four experimental sites exhibited unique snow depths, wind speeds, and air temperatures that ultimately resulted in temporal and spatial differences in *CC*. Variability was such that the greater magnitude of *CC* was not always exhibited by the top layer, but also by the middle layer (Fig. 13). For instance, in week 15, the snowpack was denser

in the top layer than in the middle layer. In week 13, the top layer snowpack was warmer than the layer beneath it. Such patterns
in temperature and density are counterexamples of the general patterns depicted in Figure 13.

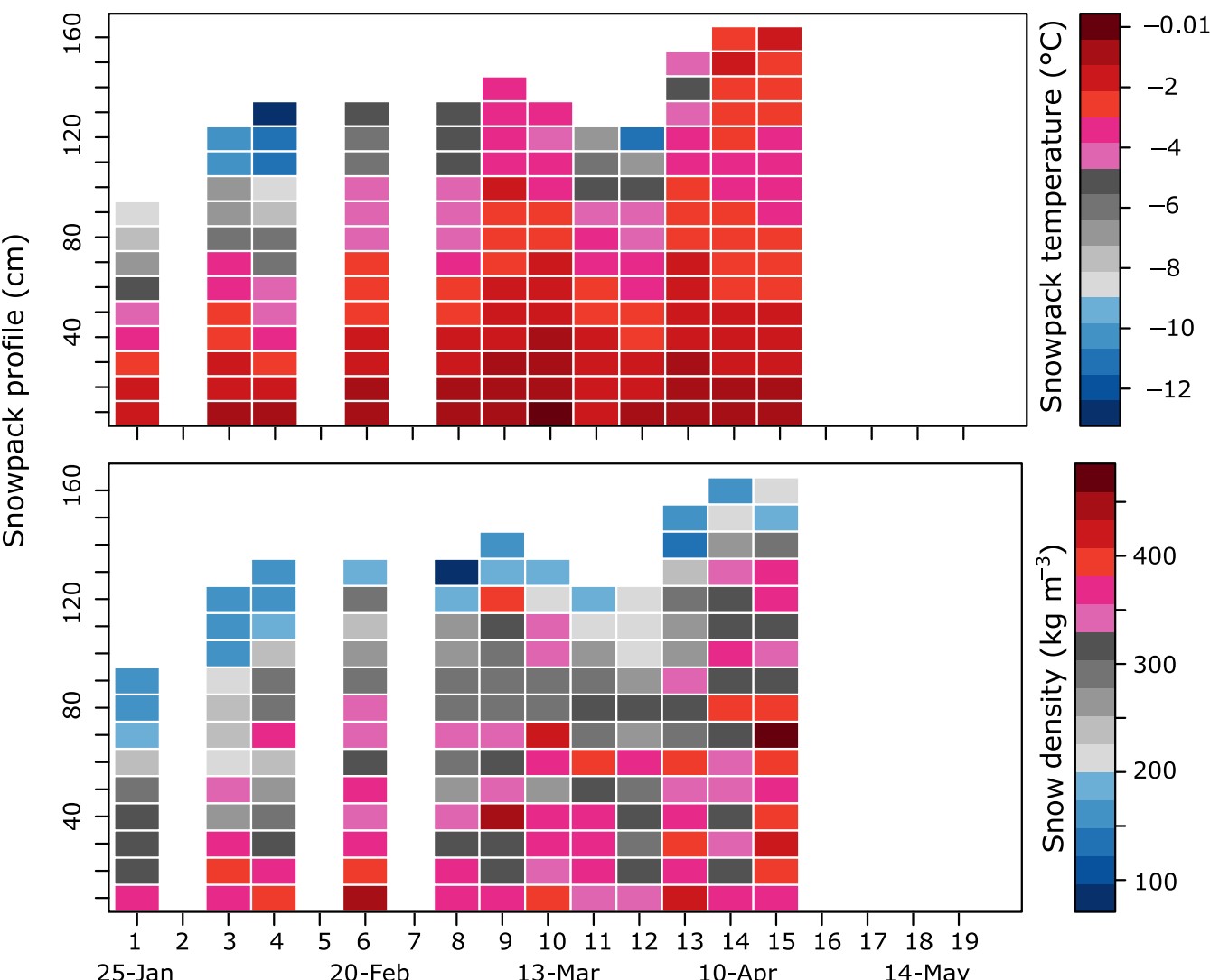

**Figure 13. Observed 10-cm snowpack temperature (top) and density (bottom) at site Sap1 from weekly snowpit surveys..**

Furthermore, the importance of snow mass on *CC* (total) at the study sites is highlighted in Figure 5. As expected, a deeper snowpack is typically associated with higher *CC*. For instance, except site Juv1, *CC* peaked at all sites in February, but more
*CC* was observed in the deeper Sap1 and Sap2 snowpacks. The same finding holds when *CC* is averaged over the 15-week period: Juv1 experienced more snow and higher *CC*, followed by Sap1. In both instances (peak and average *CC* conditions), a deeper snowpack led to larger magnitude of *CC*. In a similar study of alpine and subalpine snowpacks in the Rocky Mountains of Colorado, USA, Jennings et al. (2018) reported peak *CC* to be 2.6 times greater for the alpine snowpack than for the

subalpine location, which they mostly attributed to the higher *SWE* accumulation at the alpine site. However, Jennings et al.
(2018) also noted that colder temperatures (up to 4°C) led to higher *CC* at their alpine site.

In early February (during peak *CC* conditions), the snow depth difference between sites Sap1 and Juv1 was very small (Table 3). Nonetheless, Sap1 exhibited higher *CC* than Juv1 (Fig. 5). This is because in addition to snow depth, *CC* values depend on the density and temperature of the snow (Fig. 12). The higher peak *CC* found at site Sap1 can be explained by the higher snow density that is typically associated with higher wind velocities (Vionnet et al., 2012) and wind speed-induced densification.
As illustrated in Figure 3, site Sap1 was windier than Juv1. This is expected, as it is well known that wind speed is low within forest canopies (Davis et al., 1997; Harding and Pomeroy, 1996), such as those in site Juv1.

## 4.2 CLASS performance

Gaps between weekly snowpit surveys failed to capture short-lived events such as warm and cold spells or rain-on-snow events. In an attempt to produce higher frequency *CC* time series, we used the CLASS land surface model to simulate 30-min
bulk snow density and *SWE*. Based on our findings, CLASS successfully estimated snow density and *CC*. Although CLASS reasonably predicted *SWE*, one cannot deny the fact that it underestimated observations (Fig. 6).

Alves et al. (2020), who operated CLASS on the same experimental watershed as in this study, reported an overestimation in the upper quantile of the latent heat flux. Interestingly, a similar pattern was reported for other Canadian boreal forest sites. Similarly, Parajuli et al. (2020a) explored a simple temperature-index (TI) model, again at the same sites. They found that the
inclusion of snow sublimation led to improvements in their model performance. Based on these recent studies, it seems fair to conclude that the inadequate estimation of latent heat flux by CLASS could favor *SWE* underestimation. The precipitation biases reported in the methodology section could also have impacted *SWE* estimation. It is equally important to point out that the single-layer representation of snowpack processes by CLASS stands out as a major shortcoming. Given the limitations of bulk estimations, which are often too broad to properly describe all snowpack processes (Roy et al., 2013), several studies
have opted for multilayer snow models (Brun et al., 1997; Lehning et al., 2002; Vionnet et al., 2012). Whether the model is single- or multi-layer, certain degree of uncertainty will persist when modelling snowpack processes (Jennings et al., 2018; Raleigh et al., 2015; Alves et al., 2020). Given the prevalence of biases in the snow modelling chain, we feel confident enough to use CLASS-estimated *SWE* to derive multilayer snow density.

## 4.3 Reconstructed *CC* time-series

For *CC* reconstruction this study explored the (simpler) hybrid procedure proposed by Andreadis et al. (2009). Using this method, we generated a reasonable snow density (10-cm vertical layers) values to support the derivation of multilayer *CC* time series that are more prone to capturing short-lived events (Fig. 9). To better visualize the variability of snow density and cold content estimates, we aggregated the snowpack (10-cm layer) into top, middle and bottom snow layers (Fig. 8). One should keep in mind though that all results presented in section 3.3.2 are based on 10-cm vertical slices. As shown in Figure 7, snow
density was less well modelled for the top layer than for the other two layers, which obviously also affected the *CC* estimates

(Fig. 8). One of the challenges of snow modelling is the estimation of fresh snow density. Russell et al. (2020) explored a range of fresh snow density formulations and concluded that a constant value of 100 kg m$^{-3}$ provided a better outcome than most empirical formulations. In Fig. 14, we compare observations of the top 10 cm snow density versus model outputs from three common empirical methods: Diamond-Lowry (Russell et al., 2020), Hedstrom-Pomeroy (Hedstrom and Pomeroy, 1998),

and Brun (Shrestha et al., 2010; Vionnet et al., 2012) (Fig. 12). As proposed in Russel et al. (2020), we also explored a constant fresh snow density of 100 kg m$^{-3}$. Note that empirical methods are used to derive fresh snow density only, and in cases where the observations are taken more than 24 h after a precipitation event (Figure 14b), snow metamorphism must be taken into account in order to have a fair comparison between models and observations. Thus, to account for snow metamorphism, we resorted to Equation 5 and then derived snow density.

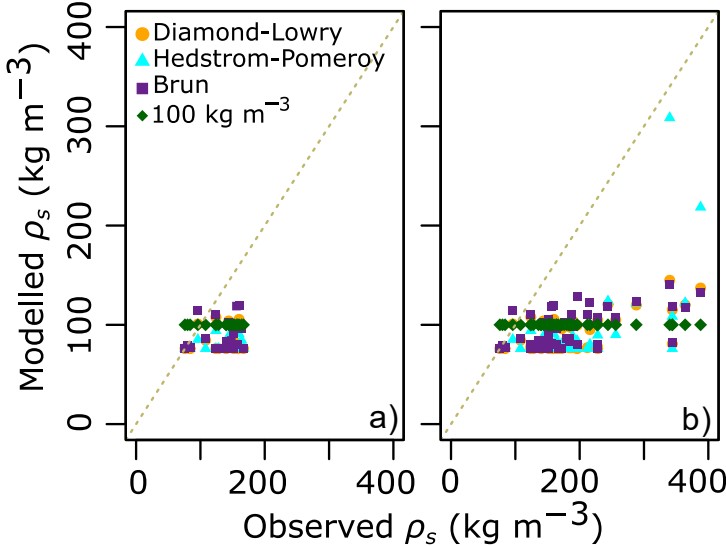


**Figure 14: Top 10-cm snow density derived from different empirical methods. a) Observed versus modelled fresh snow density (< 24 h since last snowfall); b) observed versus modelled snow density including both fresh and snow density after metamorphism based on Equation 5.**

All four methods performed poorly (Diamond-Lowry [Pbias = −53.1% and R$^2$ = 0.23], Hedstrom-Pomeroy [Pbias = −51.6%

and R$^2$= 0.25], Brun [Pbias = −50.2% and R$^2$ = 0.18], and constant 100 kg m$^{-3}$ [Pbias = −48.8% and R$^2$ = 0.11]), largely underestimating snow density (Fig. 8b). One possible cause of this underestimation is related to the presence of a canopy. It is well known that intercepted snow can stay in place for several days to months (Dewalle and Rango, 2008). Such snow can densify within the canopy and eventually unload, thereby transforming the top-layer density underneath. In agreement with this hypothesis, as noted in Figures 7 and 8, top snow density estimated at the site with the shortest vegetation (Sap1), where

there is little interception, was better than at the other sites. In a slightly different context, Raleigh and Small (2017) concluded that snow density modelling was a major source of uncertainty when studying catchment *SWE* derived from remotely sensed data.

The canopy not only intercepts some of the precipitation, but it also acts as a buffer on energy exchanges between the snowpack and the atmosphere. Indeed, the site with the shortest vegetation, Sap1, has the largest peak $CC$ (Table 3 and Fig. 9), even if it is not the site with the deepest snow cover. Indeed, looking at the whole winter, we note that site Juv1 experienced more snow and a greater magnitude of total $CC$ than Sap1 (Fig. 10). Delving into the reasons for this difference between the two sites, we find that the absence of a well-defined forest canopy appears to lead to a more responsive snow cover to meteorological forcing at site Sap1. The prevalence of higher wind speeds at site Sap1 intensifies turbulent sensible heat fluxes and thus favors the loss of heat from the snow cover in cold periods such as the one corresponding to the peak $CC$. Yet, this snow cover responsiveness at the Sap1 site does not guarantee that it has the greatest total mean $CC$, here observed at the site Juv1. Indeed, if winds increase the sensible heat flux at Sap1, they also favour the lateral transport of snow. The absence of a well-defined at canopy also means greater incoming shortwave radiation. Indeed, our CLASS simulations reveal that the average of net shortwave radiation was greater by 4.6 W m$^{-2}$ at Sap1 than at Juv1. Thus, wind scoured thinning combined with radiation enhanced ablation resulted less snow accumulation at site Sap1, and as such, a smaller mean total $CC$ ($-1.7$ MJ m$^{-2}$ vs $-1.8$ MJ m$^{-2}$ at site Juv1, see Fig. 9). This is also the reason why there were 60% occurrence where magnitude of $CC$ was higher at site Juv1 than at Sap1.

For site Mat1, there were 32% occurrences where magnitude of $CC$ were higher than at Sap1, beginning in early February and continuing through the rest of the study period (Fig. 10). Most of the time, the measured snow depth at site Mat1 was also shallower than at site Sap1. We hypothesize that cold air pooling might explain this phenomenon. During stable atmospheric boundary layer conditions, with weak synoptic forcing, there is reduced wind flows. This results in thermal decoupling in the valley depression, which favours the formation of a cold air pool (Fujita et al., 2010; Mott et al., 2016). This is substantiated by the rapid cooling of near-surface air within the valley depression, typically at night or early in the morning (Smith et al., 2010). Here, in Figure 3, we assumed that the reduced wind speed coincides with the rapidly cooling near surface temperatures (see the shaded regions) as stable atmospheric conditions prevail. During these periods, Mat1 experienced cooler air temperature than the other sites. The mean energy available for melting or refreezing at Mat1 is also smaller than for the other sites (Fig. 11). As mentioned above, smaller melt/refreeze energy contribute to the accumulation of $CC$ or the refreezing of liquid water present in the snowpack.

The site with the tallest trees, Mat2, has the lowest mean $CC$ (Table 3 and Fig. 9). This is partly due to a lower snow height on the ground (more interception) and due to the barrier effect of the canopy on incoming radiation. Also, this site experienced very few occurrences where $CC$ was larger than at site Sap1 (1% of the data), and all of them during spring melt or rain-on-snow events. It is obvious because the rain-on-snow events contribute the addition of warm sensible heat to the snowpack (Dewalle and Rango, 2008). Any heat addition results in the elimination of some snowpack $CC$ and drives destructive metamorphism to initiate melt (Seligman et al., 2014). For these periods, the taller tree at site Mat2 intercepts more rain than at site Sap1. This is the reason why the average of available melt energy was smaller at site Mat1 by 9.7 W m$^{-2}$ (rain-on-snow) and 8.4 W m$^{-2}$ (spring melting period) than at Sap1 (Fig. 11). Less availability of melt energy translates into smaller depletion

of snowpack *CC*. Thus, only during the rain-on-snow and spring melt, one may notice more snowpack *CC* at site Mat2 than at Sap1.

Initially, based on Figures 3 and 10, snow depth and air temperature appear to influence *CC* distribution across study sites. However, observed and simulated snowpack *CC* at all sites were strongly (positively) correlated with snowpack temperature
and air temperature, and weakly correlated with the snow density and snow depth values (Figure 12). It should be noted that *CC* values at all sites only showed negative correlations with snow depth (Fig. 12). When translating the relationship between *CC* and the depth of the snowpack, one must understand that there is an increase in the magnitude of *CC* with an increase in snow depth. This is because the value of *CC* is expressed negatively while snow depth is always positive. Based on *CC* observations and the hybrid procedure, we were able to identify a relationship between the mean *CC* and the tree height (Table
3 and Fig. 9). However, we were unable to report any trends in the site-wise relationship between *CC* and the above-mentioned variables (Fig. 12). Conversely, Jennings et al. (2018) attempted to establish a relationship between *CC* development and the cumulative mean of air temperature across the alpine and sub-alpine sites in the Rocky Mountains in Colorado, USA, but were unsuccessful.

**4.4 Sources of uncertainty**

Several errors and biases could arise due to poor data quality and modelling deficiencies, thereby affecting the snowmelt models (Parajuli et al., 2020a; Raleigh et al., 2015, 2016; Rutter et al., 2009). For instance, Jennings et al. (2018) applied the SNOWPACK multilayer model and reported an overestimation in fresh-snow temperature. As reported in the present study, *CC* depends heavily on snowpack temperature (Fig. 12). Any biases arising due to inaccurate derivation of snow temperature might affect *CC* estimations. The quality of model inputs also influences model performance. For instance, precipitation inputs
extracted 4 km north from present study sites were used to drive CLASS simulations, which neglects the presence of small-scale spatial variability in precipitation. Also, sites Sap1 and Juv1 benefitted from local flux tower measurements, but such direct measurements were not available for sites Mat1 and Mat2, for which many assumptions were necessary in order to create missing input time series. This problem has also been observed in several other studies(e.g. Pomeroy et al., 2007; Qi et al., 2017). Important snowpack properties beyond just *CC*, such as thermal conductivity (Oldroyd et al., 2013) and snow
interception (Hedstrom and Pomeroy, 1998) also need to be further addressed. As mentioned in section 4.2, snow density estimation presents a considerable challenge when implementing a multilayer snowpack model (Fig. 7). Therefore, future research that utilizes the physically-based snow model and describes the internal snowpack processes should focus on improving snow density estimations.

**5 Conclusion**

The purpose of this study was to document the spatial variability of *CC* in a humid boreal forest, using detailed measurements supplemented by physically-based and empirical model outputs. The studied boreal forest is characterized by a non-uniform

stand structure that led to site-to-site variations in the 10-cm weekly observations of *CC*. Areas with lower vegetation had the highest snow accumulation and thus resulted in the largest peaks in total *CC*, while the juvenile forest experienced the highest magnitude of average *CC* over the 15 weeks.

The Canadian Land Surface Scheme model was coupled with complementary empirical formulations to construct bulk, followed by 10-cm, 30-min snow density time series. Both CLASS and the empirical formulations supplied reasonable snow density and *CC* estimates. When the latter 10-cm time series were split into three layers, the bottom and the middle layers also resulted in reasonable simulations. However, modelling of the top layer was not as successful. The constructed time series were used to illustrate the influence of phenomena that are not detectable when only snowpit data are used, such as rain-on-

snow episodes or the formation of cold air pools at the bottom of the valley.

We used the Pearson's correlation coefficient (*r*) to identify the role of pertinent variables (snow density, snowpack temperature, snow depth and air temperature) that affect the distribution of *CC* at our boreal forest sites. Snowpack and air temperature appeared to be highly influential on *CC* distribution compared to the depth and the density of the snowpack. Our study was supported by 30-min time step time series of 10-cm snow temperature profiles and bias-corrected precipitation

inputs. The inclusion of such inputs helped us to reduce errors and biases. This study also highlighted the uncertainty associated with fresh snow density estimates when simulating physically-based snowmelt models.

*Data availability*: The data that support the findings in this study are available upon request to the main author.

*Author contributions*: AP and DFN (occasionally) extracted the data from the field. AP, DFN and FA designed the study. AP wrote the manuscript and analyzed the data. DFN and FA provided constructive feedback to improve the quality of the manuscript. MA performed CLASS simulations.

*Conflicts of interest*: Authors declare no conflicts of interest.


*Acknowledgements*: This research is a part of the EVAP project which was funded by Natural Sciences and Engineering Council of Canada (NSERC), Ouranos (Consortium on Regional Climatology and Adaptation to Climate Change), Hydro-Québec, Environment and Climate Change Canada, and Ministère de l'Environnement et de la Lutte aux changements climatiques through grant RDCPJ-477125-14. We would like to express our sincere gratitude to Professor Sylvain Jutras for

kindly providing the necessary sensors and support for managing the logistics of the field experiment. We conducted two successful winter campaigns, thanks to the generous support from Professor André Desrochers, who provided a snowmobile. The authors would like to thank François Larochelle and Martine Lapointe for providing some of the equipment and helping in the design of the snow profiling stations. Our study would have been incomplete without support from Annie-Claude Parent who participated in the harsh winter field campaign, managed the necessary logistics and helped conduct the field work. We

would like to express our deepest gratitude to Benjamin Bouchard and Médéric Girard, who assisted in data collection. The

authors are indebted to the Montmorency Forest staff, including Robert Côté and Charles Villeneuve, who provided generous support for managing the logistics throughout the study area. Thanks to group members Bram Hadiwijaya, Pierre-Erik Isabelle, Oliver S. Schilling, Judith Fournier, Alicia Talbot Lanciault, Georg Lackner, Carine Poncelet, Amandine Pierre, Guillaume Hazemann, Marco Alves, Adrien Pierre, Antoine Thiboult, and interns Fabien Gaillard Blancard, Jonas Götte, Kelly Proteau, and PEGEAUX members, who participated in the challenging winter field campaign.

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
