# Peer review of "Multilayer observation and estimation of the snowpack cold content in a humid boreal coniferous forest of eastern Canada"

_The Cryosphere, 2021_

## Author Comment (AC1)

1) In this study, the authors have employed a combination of empirical modelling and measurements to assess the evolution of the cold content of the snowpack in four closely located boreal forests in Québec, Canada. The general methodology of the study is explained sufficiently well and the results are of interest to the forest hydrology and water resource communities, as well as the land surface and snow modelling communities. I feel that the results are worthy of publication, but I would like to see a little more effort towards examining the differences between the snowpack properties and CC estimates between the sites.

We would like to thank the reviewer for investing his/her valuable time and effort in providing constructive feedbacks.

**Specific Comments:**

2) In Figure 5, can the authors infer from the detailed snow pit data whether the underestimated SWE is caused by excessive sublimation or melt in the model or underestimated precipitation? Are the latent heat flux measurements of any help here at two sites?

Thank you for pointing that out. We will add a new section 4.2 entitled "Performance of CLASS" to our manuscript, where we will discuss some possible reasons explaining the underestimation of SWE at the sites of interest. As mentioned, the latent heat flux overestimation by CLASS (Alves et al., 2020), precipitation bias due to distance between our sites and the precipitation gauge ($\approx$4 km) and deficiency of the single-layer snow layer scheme used in CLASS will be discussed as additional probable reasons of underestimation.

3) In Figure 6, the density of the top layer may be underestimated if the density of fresh falling snow is underestimated, or if densified snow falling from the canopy is not accounted for. If density is underestimated then CC will be too small even if temperature is correct. Site 1 which has a larger fraction of the canopy buried shows a smaller underestimation of density, suggesting that canopy effects are stronger at Sites 2, 3 and 4.

We fully agree with the reviewer. We will include a discussion related to densified snow falling beneath the canopy.

4) Would Figures 7 and 10 be more informative if small x-y plots were employed, with the $r^2$ or bias value shown? This would also provide the ability to show nonlinear or curvilinear relationships.

Thank you for this remark. To visualize the results presented in Figure 7 as a scatter plot, one can refer to Figure 6. For Figure 10, we went through the requested exercise and prepared Figure R1. The substantial amount of data for the bottom panel (reconstructed CC) makes the plot somewhat unclear, so we find it best not to include it in the paper.

[Figure]

Figure R1: Relationship between *CC* and some pertinent variables, where $\rho_s$ is snow density, *Ts* is snowpack temperature, *Ta* is air temperature and *HS* is snow depth. The top panel applies to snow pit observations and the bottom panel to the reconstructed *CC*.

5) Section 4.2: Site A1 is more exposed to turbulence, solar radiation and longwave losses, and so probably has more rapid temperature changes as weather systems change. I believe the authors are capable of adding more interpretation here. Can they infer snowfall, sublimation, decoupling, and canopy effects? I like the discussion about Site A3 on lines 334-341. I see times in Figure 2 when site A3 has the coldest temperatures. Can the authors relate this to periods with stable conditions and/or low wind speeds to bolster this discussion?

Site A1 is indeed different from the other ones as it lacks a well-defined canopy that acts as a buffer on energy and mass exchanges. Therefore, a more reactive snowpack can be expected. As pointed out by the reviewer, we will attempt to incorporate the effect of snow interception, sublimation, turbulent transport of snow, and snow accumulation and ablation pattern at our sites. In the next version of the manuscript, we include a more detailed interpretation of these processes and their impact on the *CC* at site A1.

Also, regarding cold air pooling at site A3, we will propose a new version of Figure 2 in which periods of low wind speed will be highlighted (see Figure R2 below). Also, we will add some supporting

analysis relating the effect of such short-lived events in terms of energy budget perspective.

[Figure]

Figure R2. (a) Daily 2-m air temperature ($T_a$) observations for all the study sites. (b) Daily 2-m wind speed ($u_m$) for sites A1 (sapling) and A2 (juvenile). Shaded regions denote periods of low wind speeds (< 0.8 m s$^{-1}$). Coloured points illustrate site specific snowpit surveys while period between two grey dots indicates the extensive snowpit measurement period. Spring melt started on 21 April, 2018.

6) Figure 12 and related discussion: Are the authors comparing the formulae for the density of fresh falling snow against the density found in the top 10 cm of the snowpack? I would only consider this a valid comparison if the snow survey were conducted immediately after a snowfall event and before unloading or density changes had taken place.

Sorry for the misunderstanding; we are not comparing the fresh density but deriving the superficial snow density (top 10-cm) resorting to different methods. For the sake of clarity, we have now prepared a new figure that highlights both fresh density and top snow density after metamorphism (Fig. R3). The method implemented here is:

- If there is snowfall, we apply different empirical equations, namely Diamond-Lowry (Russel et al.,2020), Hedstrom-Pomeroy (Hedstrom and Pomeroy 1998), Brun (Brun et al., 1989) and a constant 100 kg m$^{-3}$, to estimate fresh snow density.
- For every snowfall event, the top snow density is reset to fresh snow density estimate determined by one of the above empirical equations.

- If there is no snowfall, then snow undergoes metamorphism. We utilize Equation 4 to derive snow density after metamorphism.
- This is the top or superficial snow density which includes both fresh (< 24 h since last snowfall) and densified snow.
- As the top (10-cm) layer has no overlying weight, which causes densification, so we do not use Equations 5 and 6.

[Figure]

Figure R3: Top 10-cm snow density derived from different empirical methods. a) Observed versus Modelled fresh snow density (< 24 h since last snowfall) a) Observed versus modelled snow density including both fresh and snow density after metamorphism utilizing Equation 4.

Table R1: Efficiency metric for snow density derived from different empirical methods. Here $R^2$ represents the coefficient of determination and the Pbias (%) denotes the percent bias between modelled and observed snow density.

| Models | Fresh snow density | | Snow density after metamorphism | |
|---|---|---|---|---|
| | $R^2$ | Pbias (%) | $R^2$ | Pbias (%) |
| Diamond-Lowry | 0.018 | −35.2 | 0.23 | −53.1 |
| Hedstrom-Pomeroy | 0.012 | −39.6 | 0.25 | −51.6 |
| Brun | 0.028 | −32.1 | 0.18 | −50.2 |
| 100 (kg m$^{-3}$) | 5e$^{-5}$ | −24.0 | 0.11 | −48.8 |

7) Line 364-6: If the snow density estimates were disastrous when employing the Brun model, why did the authors not change to a different model or to fixed values of 100 kg/m$^3$ for fresh snow?

This is indeed a reasonable estimate for the density of fresh snow density (Fig. R3a), but not in the presence of metamorphism (Fig. R3b).

**Minor points and Corrections:**

8) Line 50: I would phrase this as "plays a central role in the timing of snowmelt". "Delaying" makes it seem like the snow is not melting at the correct time.

We will rephrase it, thank you.

9) Line 59: I would replace "resorted to" with "employed". "Resorted to" implies that snow pits are not a good method. They are labour intensive, but can provide much information.

We agree with the reviewer and will correct it.

10) Line 62-64: Is "Of note, a slight contrast was observed by Seligman et al. (2014), who reported that the contribution of spring snow storms to CC had a smaller impact on delaying snowmelt than the porous space from dry fresh snow" intended to mean, that in spring storms the snow is near 0°C and so adds the minimum possible heat content based on its mass, whereas the pore space in cold dry low density snow results in a low thermal conductivity which delays snowmelt more than the cold content of the warm spring snow? If so, a few extra words would make that more clear.

The reviewer understood the meaning of the sentence, but we will indeed add a sentence to clarify your concern.

11) Line 64-65: "However, Jennings et al. (2018) reported shifts in the onset of snowmelt by 5.7 h and 6.7 h at alpine and subalpine sites, respectively, when CC at 6:00AM was less than 0 MJ m−2." Shifts from when to when, caused by what? By definition, CC must be ≤ 0 MJ m−2 but surely the amount would affect the timing of snowmelt.

The reviewer has a good point, and we will rephrase this sentence as:

" Nonetheless, Jennings et al. (2018) tested the role of $CC$ on delaying the snowmelt and reported shifts in the onset of snowmelt by 2.3 h and 2.8 h at alpine and subalpine sites, respectively, after the removal of CC at 6:00AM . However, when the $CC$ at 6:00 AM was less than 0 MJ $m^{-2}$, there was a delay of snowmelt onset by 5.7 h at the alpine and 6.7 h at the sub-alpine site, respectively."

12) Line 81: I would delete "our" unless the field observations were provided to a different group.

We will delete it, thanks.

13) Line 83: I would change "resort to" to "employ" or "use".

Thanks, we will rectify it.

14) Line 91: I would change "forested" to "forest".

We will modify it, thanks.

15) Line 93: I would change "was" to "were".

We will change "was" to "were".

16) Line 136: I would change "enabling the stability of the prognostic modelled variables" to "ensuring an uninterrupted time series of the prognostic variables".

We will rephrase the sentence relying on reviewer's suggestion.

17) Line 152: Change "weighting" to "weighing".

Thanks, we will rectify this.

18) Line 152-3: I would acknowledge that there can be significant differences in snowfall for some snowfall events over a 4 km distance. It is probably not a consistent spatial bias, however, so the methodology is acceptable as long as there is such an acknowledgement.

We agree with the reviewer and will add some sentences to acknowledge the bias present in the precipitation input.

19) Figure 2: Could site A1 be a brighter blue to aid in seeing it as distinct from site A4? The dots are otherwise fine as long as none are completely hidden. If any are completely hidden, perhaps some can have no fill or have the sizes made smaller.

Thank you for relevant suggestion, we will change the color of the plot throughout the manuscript.

20) Figure 11: I am not sure what the light blue shading in the caption is referring to.

Thanks, we will remove the light blue shading.

Reference

Alves, M., Nadeau, D. F., Music, B., Anctil, F. and Parajuli, A.: On the performance of the Canadian Land Surface Scheme driven by the ERA5 reanalysis over the Canadian boreal forest, J. Hydrometeorol., 21(6), 1383–1404, doi:10.1175/jhm-d-19-0172.1, 2020.

Brun, E., Martin, E., Simon, V., Gendre, C. and Coleou, C.: An energy and mass model of snow cover suitable for operational avalanche forecasting, J. Glaciol., 35(121), 333–342, doi:10.1017/S0022143000009254, 1989.

Hedstrom, N. R. and Pomeroy, J. W.: Measurements and modelling of snow interception in the boreal forest, Hydrol. Process., 12(10–11), 1611–1625, doi: 10.1002/(SICI)1099-1085(199808/09)12:10/11<1611::AID-HYP684>3.0.CO;2-4, 1998.

Jennings, K. S., Kittel, T. G. F. and Molotch, N. P.: Observations and simulations of the seasonal evolution of snowpack cold content and its relation to snowmelt and the snowpack energy budget, Cryosph., 12(5), 1595–1614, doi:10.5194/tc-12-1595-2018, 2018.

Russell, M., Eitel, J. U. H., Maguire, A. J. and Link, T. E.: Toward a novel laser-based approach for estimating snow interception, Remote Sens., 12(7), 1–11, doi:10.3390/rs12071146, 2020.

---

## Author Comment (AC2)

The authors detail the evolution of snowpack cold content at 4 sites with differing vegetation cover properties in Quebec, Canada. They do this using a combination of field observations and model output over one field season at their research site. They detail differences in cold content across the sites and present extensive model validation plots and figures. There is a sufficient amount of in-depth analysis and it's clear the authors have devoted a great deal of time and care to the plots, tables, and manuscript. However, there are two major shortcomings/revisions that I feel need improvement before acceptance should be considered:

Dear Dr. Keith Jennings,

Thank you for the kind words and constructive comments. We are very confident that they will help us to build a new version with a much-improved interpretation of the results and a more straightforward take-home message.

1) Given the large number of multi-layer models available, why use a single-layer model and then attempt to split the layers?

First, it is important to recall why we need to use a snow model. As shown in equation 1 of the paper, the cold content of the snowpack is computed from snow temperature, snow density, and snow depth. During our snow pit surveys, we measured these variables jointly and calculated the associated *CC*. These results are presented in section 3 of the article.

However, several phenomena can occur between the snow pits, taken about every week. Thus, it is interesting to explore *CC* at a finer temporal scale. Thanks to our snow profiling stations, we dispose of 30-min measurements of snow temperature taken every 10-cm in the vertical direction, as well as snow depth. However, we do not have continuous measurements of density, and it is here that we decide to use a model.

Before using a model, it is essential to know its strengths and weaknesses. We chose CLASS is that it was evaluated at our site for its ability to reproduce the observed snow heights (see Alves et al., 2020). The model showed its ability to reproduce the longevity of the snowpack but tended to underestimate the snow height for the reasons discussed in Reviewer 1's comment 1.

There are, of course, several other snow models available, some of which are multi-layered, but before resorting to them, we would have had to do a complete validation at our site, which we felt was beyond the scope of the study.

Since CLASS is a single-layer model, we employ a method to generate multiple density layers, much like Roy et al. did in 2013 to estimate the SSA. Note our process of densifying each layer taken from Shrestha et al. (2010) is similar to what is implemented in the snowpack model.

We, therefore, plan to clarify the reason for our use of CLASS in the new version of the paper by synthesizing the above points.

2) It's particularly unclear to me how the Andreadis method is applied.

In agreement with the other remarks along these lines, we propose to add Figure R4, which will clarify the methodology used to obtain the reconstructed *CC* time series. It shows where the application of Andreadis (2009) method fits into the process.

[Figure]

Figure R4: Schematic diagram showcasing the methodology adopted to conduct this study.

3) The results make it seem like the authors only track 3 layers, but then subsequent sections break out layers in 10 cm increments. Do the layers stay consistent or are they combined and divided as would be done in a multi-layer model?

We agree with the reviewer that the methodology section was not sufficiently described in previous version of manuscript. Indeed, we performed the 10-cm vertical layer scheme for our analysis and then divided them into 3 layers only to visualize and understand the difference of snow density across the snowpack.

4) The methods text says simulated SWE from CLASS is used, but then the equations are all for density.

Thank you for your relevant suggestion. In our previous version, *Wns* and *Ws* (derived from CLASS) presented in Equation 6 represented the amount of new and old snow water equivalent, respectively. To make this more explicit, we will change the variable names to *SWEns* and *SWEs*, respectively.

5) Do the final simulated CC series use the thermocouple data or snowpack temperature output from CLASS? If the latter, how is temperature reallocated?

We used thermocouple data, as shown in the new Figure R4.

6) I think a schematic showing the complete workflow for the reconstructed CC time series would benefit the readers, in addition to providing much more detail in the methods text.

This is a great idea, thanks and see previous comments.

7) Scientific contribution. I'm left wondering what the major novel contribution of the paper is, which means it needs further work to be accepted in The Cryosphere.

The main original aspect of the work is as follows:

**Research topic**

As detailed in Jennings et al. (2018), the lack of direct observation still hinders the understanding of snowpack cold content. To the best of our knowledge, only a few studies have investigated this state variable. In the past, some observational studies (Seligman et al., 2014; Jennings et al., 2018) and snowmelt models have (Jost et al., 2012; Valery et al., 2014) either detailed or inserted snowpack cold content. However, only few of them attempted to understand this state variable at forest (e.g. Jost et al.,2012; Jennings et al., 2018). But none of them compared the variability of *CC* across different stand structures. In this study, we attempted to understand the *CC* variability across four contrasting sites with canopy. We believe, such comparison is first of its kind particularly for cold content studies.

**Methodology**

Aided by snow profiling station and labor-intensive field measurement, our study used a time series of snow temperature profiles (10-cm apart) and combined with an empirical formulation to produce a multi-layer cold content time series across four different forest sites. The manual weekly observation of multi-layer snow density, snowpack temperature, and snow depth enabled us to understand the behavior of *CC* across our forest sites. The use of 30 minutes 10-cm snowpack temperature profiles combined with snow depth and empirical snow density estimate provided us with a multi-layer time series of *CC*. We believe using an automated snowpack temperature profile at four sites to derive *CC* is the first of its kind as well.

**Take-home message**

Our study aimed to understand the *CC* behavior across four different forest stands. Based on our findings, we were able to relate the role of vegetation on the accumulation and ablation of snow and its role on *CC*. Furthermore, we have presented the effect of short-lived events, such as the effect of cold air pooling on *CC*. Here site A3 exhibited a unique thermal regime which ultimately resulted in the difference of *CC* at this site compared to others. We also would like to thank the reviewer for raising pertinent issues related to the inclusion of the rain-on-snow event to bolster our discussion. Indeed, the rain-on-snow event resulted in *CC* removal and led to almost uniform *CC* distribution across our sites (regardless of differences in vegetation and vegetation altered energy exchanges).

As presented in Jennings et al. (2018), the authors detailed the relationship between cumulative temperature and precipitation with cold content development. However, due to snow interception, snow accumulation and melt are more variable in forest than any other locations (Parajuli et al.,2020). In a forest, the intercepted snow may immediately unload or stay there (densify) for several days to months and unload (Dewalle and Rango, 2008). Thus, this study attempted to relate pertinent variables such as snow depth, air and snowpack temperature, and snow density. We believe this is also a novel aspect of our study.

8) Overall, my feeling is the authors can devote less results text to model validation and output (particularly the analysis of the 3-layer scheme as I note in my line-by-line comments below) and spend more time on process-based analysis. For example, what happens during the rain-on-snow events from an energy balance perspective? Why are there differences in cold content development across the sites? Is it the effect of canopy cover on snow accumulation and/or the snowpack energy balance? Given the field observations and modeling, I feel there is a lot more the authors could unpack that give us, the readers, deeper information on the processes governing cold content development and removal at these particular sites.

We agree with the reviewer and add several sentences to describe different energy budget components and other process-based analyses in our discussion. This includes the role of the canopy on $CC$ variability, the role of latent flux difference on snow accumulation and ablation and ultimately $CC$, the effect of rain-on-snow events on $CC$ distribution, the importance of solar radiation and its role on $CC$ variability, and the importance of snow interception on snow densification in forest and its role on $CC$. We will also compare different short-lived events from an energy balance perspective.

**Line-by-line comments:**

9) Line 9: Surface melt can begin before snowpack cold content = 0.

We will remove the sentence, thanks.

10) Line 37: The snowpack energy balance predates USACE (1956). For example, Angström's work from the 1910s on the radiation budget and Western Snow Conference papers from the 1940s published in Transactions of the American Geophysical Union. I'm sure there's plenty more going back in time.

Thanks for bringing this up, we will update the reference.

11) Line 37: DHSVM was developed with a single-layer snowpack, but now uses a two-layer formulation (Wigmosta et al., 2002): Wigmosta, M. S., Nijssen, B., Storck, P., & Lettenmaier, D. P. (2002). The distributed hydrology soil vegetation model. Mathematical models of small watershed hydrology and applications, 7-42.

We agree with the reviewer. Thanks for your concern we will update it accordingly.

12) Line 46: $\rho_w$ is not in the equation

We will rectify the error, thanks.

13) Line 55: Change snow surveys to snow pits.

We will change this, thanks.

14) Line 55: Reconsider "tedious and demanding." I'd probably say time-consuming (I find digging snow pits to be quite enjoyable, not tedious).

We will rephrase the sentence, thanks.

15) Line 56: You can note example datasets here. E.g. the Williams and Morse Niwot LTER data: Williams, M., J. Morse, and Niwot Ridge LTER. 2020. Snow cover profile data for Niwot Ridge and Green Lakes Valley, 1993 - ongoing. ver 15. Environmental Data Initiative. https://doi.org/10.6073/pasta/a5fca9d02a4a6a0744cc0d0ffccacd09 (Accessed 2021-05-06).

We will cite the reference you have provided, thanks.

16) Lines 56–59: This reads very similarly to the opening sentence of paragraph 2 in Jennings et al. (2018). Please note as such.

We will reword the text to avoid too much similarity to Jennings et al. (2018) and add a reference to this paper.

17) Line 59: Please replace all uses of resort/resorted with a more appropriate word (e.g., used, leveraged, utilized, etc.). Resort typically has a negative connotation (i.e., there was nothing else we could use so we had to use this.).

We will replace the word throughout the manuscript, thanks.

18) Lines 62–64: This was noted in Jennings et al. (2018) (please add text as such). Lines 65–66: This is again very similar to the language in the Jennings et al. (2018) manuscript. Please be careful with paraphrasing and providing citations for paraphrased work.

We will improve our paraphrasing and add the reference.

19) Line 68: Jennings et al. (2018) includes a forested site (the subalpine) and compares it to a higher, open site (the alpine).

We agree with the author and will add this precision in the paper.

20) Line 69: Remove "obviously"

We will remove it, thanks.

21) Lines 78–79: "model articulated around the energy balance" > rephrase for clarity.

We will rephrase the sentence, thanks.

22) Line 84: "even models such as this are not free of biases" > rephrase because all models have some form of bias. You could highlight the previously cited snow model comparison literature.

We will add a supporting sentence for our statement.

"For instance, Raleigh et al. (2016) tested three physically based snow models (Utah Energy Balance (UEB), Distributed Hydrology Soil Vegetation Model (DHSVM) and Snow Thermal Model SNTHERM) and reported bias in longwave radiation estimation ranging from −12 to +18 W m$^{-2}$."

23) Line 87: What is "acceptable" in this context? Most readers, myself included, will not have a good intuition of what large and small SSA values are.

Thank you for your concern. For the sake of clarity in the range of SSA value we will add the following sentences.

"For instance, Roy et al. (2013) disaggregated CLASS-derived snow water equivalents into multilayer values at each time step, for the purpose of estimating the specific surface area (SSA) of a snowpack. In their study, the authors reported a specific surface area ranging from 33.1 to 155.8 m2 kg$^{-1}$, for an acceptable root mean square error (RMSE) of 8.0 m2 kg$^{-1}$ in CLASS-derived SSA for individual layers."

24) Line 89: Remove "obvious" and rephrase sentence. I think you're referring to cold content modeling specifically, but there are many forest snow model studies.

Based on the reviewer suggestion, we will rephrase the sentence.

25) Line 105 (figure 5 caption and throughout): Consider changing the names A1 through A4 to more meaningful terms or abbreviations. There are three forest cover classes (sapling, juvenile and mature). You could use some variation thereof and specify which mature forest site is denser.

We will change the name to "Sap1", "Juv1", "Mat1" and "Mat2" instead of A1 through A4.

26) Lines 117–124: This could use greater explanation. What instruments were used, what was the vertical frequency of density and temperature sampling, etc? How were the pit values used to impute the missing data?

The snow pit data (snowpack density, temperature, and depth) are exclusively used for the observational study (sections 3.1, 3.2, and 4.1). For model validation, we used the snow density from snow pit surveys. However, the snow profiling station provided the 30-minute 10-cm snowpack temperatures.

27) Lines 128–130: "A simple approach would be to interpolate the density values extracted from snowpits, but this would be incomplete and error-prone given their limited number and absence early in the season." This seems overly subjective, especially considering the low $r^2$ values for the hybrid approach.

Our site has the particularity of being very snowy, and it is not uncommon for episodes delivering >10 cm of fresh snow to occur (see Figure 8 from the paper). In this sense, a simple linear interpolation of density values appears to be a hazardous exercise. Also, as stated in the cited sentence, if we were to do this, we would have no value during the early season and very few towards the end of the melting period. This will be emphasized in the next version of the paper.

28) Lines 136–137: Further explanation needed. For example, what are the prognostic variables?

We thank reviewer for raising this issue. We will add the prognostic variables.

"In this analysis, CLASS version 3.6 was used in offline mode and with a 30-min time step, ensuring an uninterrupted time series of the prognostic variables (Roy et al., 2013) thereby allowing the inclusion of multiple soil layers and accounting for snow interception, snow thermal conductivity, and snow albedo, as described in Bartlett et al. (2006)."

29) Line 147: Change Table 2 to Table 1

Thank you for your suggestion. We will change this.

30) Table 2: It doesn't seem like all sites should be marked as having PPT measurements when the text says it was measured at one location 4km away?

We agree with the reviewer and will modify the table accordingly, thanks.

31) Line 165 (Section 2.2.3): Please see my major revision comment #1. This section needs significant improvement in terms of clarity and specificity.

Thank you for raising this issue. We have prepared a schematic diagram (Fig. R4) and will take great caution at explaining every steps of the approach.

32) Lines 169–171: Text says minimum, but equation uses maximum.

Indeed, the text says minimum as we took the minimum value of snow density derived by snow pit surveys which is 76 kg m⁻³. However, the use of Equation 2 sometimes resulted in snow density estimates below 76 kg m⁻³. Thus, we used a maximum value as in Brun et al. (1989).

33) Lines 206–207: Are these differences computed within the snowpack at each site per time or computed across all sites?

Here, we presented the minimum and maximum cold content from all the sites for individual layer. We will clarify this in the next version.

34) Line 213 (and throughout text): Please change amplitude to magnitude.

We will change them throughout the manuscript, thanks.

35) Figure 3: I like the amount of info conveyed by this figure, but find the color scale to be confusing. Because it's representing a single, non-divergent variable, you would be better suited by using the shade of single color. Also, please add "Layer cold content" to the scale bar.

The reviewer raised a good point. We have prepared a new version of the plot with shades of a single colour (Fig. R5). We find it more difficult to make a quantitative interpretation of the figure and

would prefer to keep the original version.

[Figure]

Figure R5: Example of multi-layer cold content observation using the shades of the purple colour.

36) Line 222: Is this the maximum difference or difference in maximum CC? Table 3 shows a larger difference between A1 and A4.

Indeed, here we are referring to the mean total *CC*. As such, we believe this statement is correct.

37) Table 3: The minimum appears to occur in late March (early April?) for site A2 as seen in Fig 4.

True. We will correct this mistake, thanks.

38) Line 232: Please use less subjective terms (for example, "CLASS produces low mean biases in...").

Thanks, we will modify the sentence.

39) Line 233: Somewhat interestingly, the negative SWE bias would indicate the model has a cold bias in snowpack temperature in order to get CC so accurately. Please include snowpack temperature in the validation.

See our answer to comment 1 from Reviewer 1.

40) Line 239: It appears you're producing a 3-layer scheme, but this is not specified in the methods section. Please add that along with information on how the layers are defined/combined/separated (see major revision comment #1).

See our answer to your comment 3.

41) Lines 240–242: You need to clarify when you're using snow temperature from the thermocouple arrays versus from the snow pit or CLASS. It's not clear here or in the methods section. I think it would be worth adding some material to the methods and again to the results so the readers are certain when observed versus simulated values are being used.

The new figure (Fig. R4) will avoid any confusion in this regard.

42) Line 249: This statement seems incorrect. If you're using the snow temperature from the thermocouple, then most of the error is coming from the density estimates and layering scheme.

True, most of the error is coming from density estimate. Thus, we have portrayed Figure 11, to identify the probable reason for uncertainty in snow density estimate.

43) Figure 6 and preceding text: I'm leaning towards getting rid of the arbitrary 3-layer scheme and only validating/describing the 10 cm layer results shown below. It's unclear what information/utility the 3-layer scheme provides given that most of the findings are based on the observations and the 10 cm layer discretization.

It is a way to visualize a large amount of data. To reproduce the same figure but for 10-cm slices would be very difficult to analyze in our opinion.

44) Figure 7: Please change to a table. The color scale provides the same information as the numbers, but the numbers on their own are easier to interpret.

We prefer to keep the plot as it is.  A similar figure is presented in Parajuli et al. (2020a) and we believe there are several advantages to it. For instance, the colour box immediately notifies the reader about the existing difference in our error metric. Also, the text will provide a minor detail present in this plot. We believe, both text and color scale are complimenting each other.

45) Lines 264–265: Were the rain-on-snow events missing because no snow pits were dug at those times or the snow pits suggested different changes to cold content than the simulations? Either way, this needs to be clarified.

Indeed, no snow pits were dug during rain-on-snow events. Please refer to Line 119 for more details.

46) Figure 8: Same comment as figure 3. Consider changing the color ramp and adding "Layer cold content" to the scale bar.

Please refer to comment 35, thanks.

47) Lines 273–274: This assertion is presented without supporting evidence. In line with my major revision comment #2, this would be an ideal place to test some process-based hypotheses.

We disagree with the reviewer. Parajuli et al. (2020b) reported the effect of stand structure on snow accumulation and ablation in the present research site, which is acknowledged in the discussion section. Here, as presented in Table 1, the stand structure is not uniform. Also, differences in snow accumulation and ablation are shown in Figure 9b, influenced mainly by vegetation. I believe this

assertion is not presented without supporting evidence. However, we will add some more details to justify our statement.

48) Figure 9: Why is simulated cold content plotted with observed snow depth? It seems like observed snow depth should be plotted with earlier observational figures. Also, keep the rain-on-snow shading consistent with previous figure.

Here, our simulation is largely built from observations. We believe the new figure (Fig. R4) helps clarify this. We agree with the reviewer for the rain-on-snow shadings and will modify this accordingly.

49) Lines 281–285: These results need more unpacking and their associated methods need to be moved to the methods section. You should also be careful with "positive" and "negative" correlations here. Most cold content values have been discussed in terms of their magnitude, which may lead readers to think cold content declines (i.e. approaches 0) when air temperature decreases. Additionally, simulated and observed values need to be noted explicitly in the text along with the period of comparison (is CC correlated with 30-min air temperature, daily air temperature, or average air temperature to date?).

We will be more transparent in this section. We will move all material associated with the methodology to the appropriate section. Also, the negative correlation is due to negative values of $CC$ and positive values of snow depth. We will modify the text accordingly.

50) Lines 286–291: Similar to my above comment, this needs much further explanation. This section could become an important part of the paper (and its novel contribution to the field) if you can further evaluate forest cover differences and their quantitative effect on cold content evolution. For example, you could include an assessment of snowpack energy balance differences and/or changes in snow accumulation as caused by forest cover in both observations and the model.

We disagree with the reviewer and believe that we have evaluated the forest cover difference and their quantitative role on $CC$ evolution. Figure 3 to Figure 10 detail the spatiotemporal variability of cold content with respect to the vegetation. As illustrated in Figure 3 and Figure 4, based on observations, the detailed multi-layer cold content time series is presented for respective sites with diverse stand structures.  In Figure 9, we sought the relationship between the snow depth (accumulation and ablation of snow) and its role on $CC$. Figure 10 attempted to establish the relationship between different pertinent vegetation inputs with $CC$. In this plot, we performed such an assessment detailing differences in the vegetation. In the result section, we have described the mean, peak, and even the variability of $CC$ across sites. Nonetheless, we will improve and add more relevant information as suggested by both reviewers. We agree with the reviewer, and we will now present the $CC$ variability and short-lived events with a snowpack energy budget perspective. Please refer to comment 8.

51) Figure 10: Please change to table (same comment as figure 7).

Please refer to comment 44.

52) Line 298–299: Figure 11 does not show cold content.

Indeed, the cold content time-series for site Sap1 was plotted before (Please refer to Figure 3)

53) Figure 11: Need to clarify if these are simulated or observed values. There is no light-blue shading in the plots. Also, the color ramp should not be divergent as the values they represent are not. Consider using gradation of single color.

We agree with the reviewer and will modify accordingly, thanks. For colour ramp please see the comment above.

54) Lines 304–309: Figure 4 does not show mass, only cold content. It might be worth further evaluating SWE, depth, and cold content over time in the results sections. Also note that the average winter temperatures in Jennings et al. (2018) were ~4°C cooler at the alpine site (colder frozen mass and more of it led to greater CC in the alpine).

True, Figure 4 is based on weekly measurements, and we don't have complete series of SWE and depth. However, as presented in Figure 9, we have plotted *CC* with the snow depth as the snow-profiling stations provided continuous time series of snow depth.

Thank you for providing the detail description of Jennings et al. (2018), we will add this information.

55) Lines 318–350: ) Lines 318–350: I like this section, but I feel like the paper would have a greater impact if there was a greater reliance on results versus discussion when comparing the sites (please see my comment on lines 286–291). For example, you discuss cold air pooling here, but don't provide data. Why not add this to the results section with data from the air temperature sensors at the different sites? If these data don't support the hypothesis, then it can be removed. Data from Jennings et al. indicated that the energy balance was typically positive when snow was not actively accumulating. However, there were exceptions at night as a result of radiative cooling from the snowpack. You could provide a comparison from your sites here by providing energy balance output from CLASS in the results.

We agree with the reviewer and will modify the cold air pooling discussion. As pointed out by reviewer 1, we will add the discussion relating to the low wind speed (stable atmospheric condition) and cold air pooling mechanism, referring to Figure R2. Also, we will compare the energy balance output for the period with low wind speeds and rain-on-snow events.

56) Figure 12: This is an unfair comparison. The density of freshly fallen snow is not comparable to the density of the top 10 cm of a snowpack. There's no fresh snow I know of that falls at 400 kg m$^{-3}$.

Sorry for misunderstanding. However, the plot displays the superficial snow density not the fresh one.  Please refer to Figure R2

57Lines 362–366: Please see comment above. Consider removing plot and text unless the analysis is changed to provide a more important validation of new snow.

Please refer to above comment 56.

58) Line 397: Please add link in revised manuscript.

At this point, we are working on a data manuscript and will share our data once our paper undergoes discussion. Also, the dataset will be available from authors on a reasonable request immediately. Thanks.

Reference

Alves, M., Nadeau, D. F., Music, B., Anctil, F. and Parajuli, A.: On the performance of the Canadian Land Surface Scheme driven by the ERA5 reanalysis over the Canadian boreal forest, J. Hydrometeorol., 21(6), 1383–1404, doi:10.1175/jhm-d-19-0172.1, 2020.

Andreadis, K. M., Storck, P. and Lettenmaier, D. P.: Modeling snow accumulation and ablation processes in forested environments, Water Resour. Res., 45(5), 1–13, doi:10.1029/2008WR007042, 2009.

Bartlett, P. A., MacKay, M. D. and Verseghy, D. L.: Modified snow algorithms in the Canadian land surface scheme: Model runs and sensitivity analysis at three boreal forest stands, Atmosphere-Ocean, 44(3), 207–222, doi:10.3137/ao.440301, 2006.

Brun, E., Martin, E., Simon, V., Gendre, C. and Coleou, C.: An energy and mass model of snow cover suitable for operational avalanche forecasting, J. Glaciol., 35(121), 333–342, doi:10.1017/S0022143000009254, 1989.

DeWalle, D. R. and Rango, A.: Principles of snow hydrology, 1 st ed., Cambridge University Press, New York., 2008.

Jennings, K. S., Kittel, T. G. F. and Molotch, N. P.: Observations and simulations of the seasonal evolution of snowpack cold content and its relation to snowmelt and the snowpack energy budget, Cryosph., 12(5), 1595–1614, doi:10.5194/tc-12-1595-2018, 2018.

Jost, G., Moore, R. D., Smith, R. and Gluns, D. R.: Distributed temperature-index snowmelt modelling for forested catchments, J. Hydrol., 420–421, 87–101, doi:10.1016/j.jhydrol.2011.11.045, 2012.

Parajuli, A., Nadeau, D. F., Anctil, F., Schilling, O. S. and Jutras, S.: Does data availability constrain temperature-index snow model? A case study in the humid boreal forest, Water, 12(8), 1–22, doi:10.3390/w12082284, 2020a.

Parajuli, A., Nadeau, D. F., Anctil, F., Parent, A.-C., Bouchard, B., Girard, M. and Jutras, S.: Exploring the spatiotemporal variability of the snow water equivalent in a small boreal forest catchment through observation and modelling, Hydrol. Process., 34(11), 2628–2644, doi:10.1002/hyp.13756, 2020b.

Raleigh, M. S., Livneh, B., Lapo, K. and Lundquist, J. D.: How does availability of meteorological forcing data impact physically based snowpack simulations?, J. Hydrometeorol., 17(1), 99–120, doi:10.1175/JHM-D-14-0235.1, 2016.

Roy, A., Royer, A., Montpetit, B., Bartlett, P. A. and Langlois, A.: Snow specific surface area simulation using the one-layer snow model in the Canadian LAnd Surface Scheme (CLASS), Cryosph., 7(3), 961–975, doi:10.5194/tc-7-961-2013, 2013.

Seligman, Z. M., Harper, J. T. and Maneta, M. P.: Changes to snowpack energy state from spring storm events, Columbia River headwaters, Montana, J. Hydrometeorol., 15(1), 159–170, doi:10.1175/JHM-D-12-078.1, 2014.

Shrestha, M., Wang, L., Koike, T., Xue, Y. and Hirabayashi, Y.: Improving the snow physics of WEB-DHM and its point evaluation at the SnowMIP sites, Hydrol. Earth Syst. Sci., 14(12), 2577–2594, doi:10.5194/hess-14-2577-2010, 2010.

Valéry, A., Andréassian, V. and Perrin, C.: 'As simple as possible but not simpler': What is useful in a temperature-based snow-accounting routine? Part 2 – Sensitivity analysis of the Cemaneige snow accounting routine on 380 catchments, J. Hydrol., 517, 1176–1187, doi:10.1016/j.jhydrol.2014.04.058, 2014.

---

## Author Response (AR1)

Dear editor,
The authors would like to thank you for providing us with the opportunity to revise our paper. We would like to thank both reviewers for providing the constructive feedback and improve overall quality of our research manuscript. We have either modified or provided proper justification to the reviewer's concerns. In the track change pdf file, the newly added reference is highlighted with yellow colour. Below here is the response to the reviewer's concern and modification made to address the raised concern.

**Reviewer 1**

1) In this study, the authors have employed a combination of empirical modelling and measurements to assess the evolution of the cold content of the snowpack in four closely located boreal forests in Québec, Canada. The general methodology of the study is explained sufficiently well and the results are of interest to the forest hydrology and water resource communities, as well as the land surface and snow modelling communities. I feel that the results are worthy of publication, but I would like to see a little more effort towards examining the differences between the snowpack properties and CC estimates between the sites.

We would like to thank the reviewer for investing his/her valuable time and effort in providing constructive feedbacks.

**Specific Comments:**

2) In Figure 5, can the authors infer from the detailed snow pit data whether the underestimated SWE is caused by excessive sublimation or melt in the model or underestimated precipitation? Are the latent heat flux measurements of any help here at two sites?

Response: Thank you for pointing that out. We will add a new section 4.2 entitled "Performance of CLASS" to our manuscript, where we will discuss some possible reasons explaining the underestimation of SWE at the sites of interest. As mentioned, the latent heat flux overestimation by CLASS (Alves et al., 2020), precipitation bias due to distance between our sites and the precipitation gauge (≈4 km) and deficiency of the single-layer snow layer scheme used in CLASS will be discussed as additional probable reasons of underestimation.

Modification: We have now added a new section entitled "CLASS performance". Please refer to this section, thanks.

3) In Figure 6, the density of the top layer may be underestimated if the density of fresh falling snow is underestimated, or if densified snow falling from the canopy is not accounted for. If density is underestimated then CC will be too small even if temperature is correct. Site 1 which has a larger fraction of the canopy buried shows a smaller underestimation of density, suggesting that canopy effects are stronger at Sites 2, 3 and 4.

Response: We fully agree with the reviewer. We will include a discussion related to densified snow falling beneath the canopy.

Modification: Please refer to line 435-439 which now read:

"One possible cause of this underestimation is related to the presence of a canopy.  It is well known that intercepted snow can stay in place for several days to months (Dewalle and Rango, 2008). Such snow can densify within the canopy and eventually unload, thereby transforming the top-layer density underneath. In agreement with this hypothesis, as noted in Figures 7 and 8, top snow density estimated at the site with the shortest vegetation (Sap1), where there is little interception, was better than at the other sites."

4) Would Figures 7 and 10 be more informative if small x-y plots were employed, with the r² or bias value shown? This would also provide the ability to show nonlinear or curvilinear relationships.

Response: Thank you for this remark. To visualize the results presented in Figure 7 as a scatter plot, one can refer to Figure 6. For Figure 10, we went through the requested exercise and prepared Figure R1. The substantial amount of data for the bottom panel (reconstructed CC) makes the plot somewhat unclear, so we find it best not to include it in the paper.

[Figure]

Figure R1: Relationship between CC and some pertinent variables, where $\rho_s$ is snow density, Ts is snowpack temperature, Ta is air temperature and HS is snow depth. The top panel applies to snow pit observations and the bottom panel to the reconstructed CC.

Modification: No modification has been made in this comment.

5) Section 4.2: Site A1 is more exposed to turbulence, solar radiation and longwave losses, and so probably has more rapid temperature changes as weather systems change. I believe the authors are capable of adding more interpretation here. Can they infer snowfall, sublimation, decoupling, and canopy effects? I like the discussion about Site A3 on lines 334-341. I see times in Figure 2 when site A3 has the coldest temperatures. Can the authors relate this to periods with stable conditions and/or low wind speeds to bolster this discussion?

Response: Site Sap1 is indeed different from the other ones as it lacks a well-defined canopy that acts as a buffer on energy and mass exchanges. Therefore, a more reactive snowpack can be expected. As pointed out by the reviewer, we will attempt to incorporate the effect of snow interception, sublimation, turbulent transport of snow, and snow accumulation and ablation pattern at our sites. In the next version of the manuscript, we include a more detailed interpretation of these processes and their impact on the CC at site Sap1. Also, regarding cold air pooling at site Mat1, we will modify Figure 3 in which periods of low wind speed will be highlighted. Furthermore, we will add some supporting analysis relating the effect of such short-lived events in terms of energy balance perspective.

Modification: As suggested by reviewer, we have now added several sentences to address the issue raised by reviewer 1. Please refer to line 442 – 475 for more detail.

Also, regarding cold air pooling at site Mat1, we propose a new version of Figure 3 in which periods of low wind speed will be highlighted (see Figure R2 below).

[Figure]

Figure R2. (a) Daily 2-m air temperature (*Ta*) observations for all the study sites. (b) Daily 2-m wind speed (*u$_m$*) for sites A1 (sapling) and A2 (juvenile). Shaded regions denote periods of low wind speeds (< 0.8 m s$^{-1}$). Coloured points illustrate site specific snowpit surveys while period between two grey dots indicates the extensive snowpit measurement period. Spring melt started on 21 April, 2018.

Finally, based on both reviewers' suggestion, we have now added a new figure (Figure R3) detailing the effect of short-lived events such as rain-on-snow and the cold air pooling phenomenon in terms of energy budget perspective. Also, in the same figure we have added the difference in energy budget during accumulation and spring melting period for our sites of interest.

[Figure]

Figure R3: Available melting/freezing energy for our sites of interest. Here WA denotes the winter accumulation period, SM the spring melting period, RS the rain-on-snow events and CP the low wind speed coinciding with smaller air temperature at site Mat1.

6) Figure 12 and related discussion: Are the authors comparing the formulae for the density of fresh falling snow against the density found in the top 10 cm of the snowpack? I would only consider this a valid comparison if the snow survey were conducted immediately after a snowfall event and before unloading or density changes had taken place.

Response: Sorry for the misunderstanding; we are not comparing the fresh density but deriving the superficial snow density (top 10-cm) resorting to different methods. For the sake of clarity, we have now prepared a new figure that highlights both fresh density and top snow density after metamorphism (Fig. R3). The method implemented here is:

- If there is snowfall, we apply different empirical equations, namely Diamond-Lowry (Russel et al.,2020), Hedstrom-Pomeroy (Hedstrom and Pomeroy 1998), Brun (Brun et al., 1989) and a constant 100 kg m$^{-3}$, to estimate fresh snow density.
- For every snowfall event, the top snow density is reset to fresh snow density estimate determined by one of the above empirical equations.
- If there is no snowfall, then snow undergoes metamorphism. We utilize Equation 4 to derive snow density after metamorphism.
- This is the top or superficial snow density which includes both fresh (< 24 h since last snowfall) and densified snow.
- As the top (10-cm) layer has no overlying weight, which causes densification, so we do not use Equations 5 and 6.

Modification: For the sake of the clarity, we have now added the fresh density estimate utilizing different empirical equation. Please refer to Figure R4.

[Figure]

Figure R4: Top 10-cm snow density derived from different empirical methods. a) Observed versus Modelled fresh snow density (< 24 h since last snowfall) a) Observed versus modelled snow density including both fresh and snow density after metamorphism utilizing Equation 4.

Table R1: Efficiency metric for snow density derived from different empirical methods. Here R$^2$ represents the coefficient of determination and the Pbias (%) denotes the percent bias between modelled and observed snow density.

| Models | Fresh snow density | | Snow density after metamorphism | |
|---|---|---|---|---|
| | R$^2$ | Pbias (%) | R$^2$ | Pbias (%) |
| Diamond-Lowry | 0.018 | −35.2 | 0.23 | −53.1 |
| Hedstrom-Pomeroy | 0.012 | −39.6 | 0.25 | −51.6 |
| Brun | 0.028 | −32.1 | 0.18 | −50.2 |
| 100 (kg m$^{-3}$) | 5e$^{-5}$ | −24.0 | 0.11 | −48.8 |

7) Line 364-6: If the snow density estimates were disastrous when employing the Brun model, why did the authors not change to a different model or to fixed values of 100 kg/m$^3$ for fresh snow?

Response: This is indeed a reasonable estimate for the density of fresh snow density (Fig. R4a), but not in the presence of metamorphism (Fig. R4b).

Modification: Please refer to Figure R4b, thanks.

**Minor points and Corrections:**

8) Line 50: I would phrase this as "plays a central role in the timing of snowmelt". "Delaying" makes it seem like the snow is not melting at the correct time.

Response: We will rephrase it, thank you.

Modification: The sentence now reads: "*CC* plays a central role in the timing of snowmelt (Molotch et al., 2009), as a deep, dense, and cold snowpack requires a substantial amount of energy for snow to reach 0°C and initiate melt."

9) Line 59: I would replace "resorted to" with "employed". "Resorted to" implies that snow pits are not a good method. They are labour intensive, but can provide much information.

Response: We agree with the reviewer and will correct it.

Modification: We have corrected the word. Please refer to line 64, thanks.

10) Line 62-64: Is "Of note, a slight contrast was observed by Seligman et al. (2014), who reported that the contribution of spring snow storms to CC had a smaller impact on delaying snowmelt than the porous space from dry fresh snow" intended to mean, that in spring storms the snow is near 0°C and so adds the minimum possible heat content based on its mass, whereas the pore space in cold dry low density snow results in a low thermal conductivity which delays snowmelt more than the cold content of the warm spring snow? If so, a few extra words would make that more clear.

Response: The reviewer understood the meaning of the sentence, but we will indeed add a sentence to clarify your concern.

Modification: Based on your suggestion, we have now modified the sentence which now reads:

"This delaying effect of CC on melt is less marked in spring, (Seligman et al., 2014). Indeed, during spring storms, fresh snow is near 0°C, and thus adds little cold content to the existing snow cover. Under these conditions, it is therefore the addition of a new dry interstitial space (which must reach saturation) that is primarily responsible for delaying melting. Quantitatively, Seligman et al. (2014) reported that the addition of pore spaces by dry fresh snow was responsible for 86% of the energy deficit within the snowpack of Columbia River headwaters."

11) Line 64-65: "However, Jennings et al. (2018) reported shifts in the onset of snowmelt by 5.7 h and 6.7 h at alpine and subalpine sites, respectively, when CC at 6:00AM was less than 0 MJ m$^{-2}$." Shifts

from when to when, caused by what? By definition, CC must be ≤ 0 MJ m−2 but surely the amount would affect the timing of snowmelt.

Response: The reviewer has a good point, and we will rephrase this sentence.

Modification: The sentence now reads:

"The authors found that newly fallen snow was responsible for 84.4% and 73.0% of the daily gains in *CC* for alpine and subalpine snowpacks, respectively. The authors also tested the role of *CC* in delaying snowmelt. When *CC* was zero at 6:00 AM, the onset of snowmelt was delayed on average by 2.3 and 2.8 h at the alpine and subalpine sites, respectively. However, when *CC* at 6:00 AM was less than 0 MJ m$^{-2}$, the onset was delayed by as much as 5.7 h at the alpine site and 6.7 h at the sub-alpine site."

12) Line 81: I would delete "our" unless the field observations were provided to a different group.

Response: We will delete it, thanks.

Modification: We have now deleted "our" from our sentence. Please refer to line 93, thanks.

13) Line 83: I would change "resort to" to "employ" or "use".

Response: Thanks, we will rectify it.

Modification: We have now changed "resort to" to "use"

14) Line 91: I would change "forested" to "forest".

Response: We will modify it, thanks.

Modification: We have now modified the word "forested" to "forest". Please refer to line 105.

15) Line 93: I would change "was" to "were".

Response: We will change "was" to "were".

Modification: Done, thanks.

16) Line 136: I would change "enabling the stability of the prognostic modelled variables" to "ensuring an uninterrupted time series of the prognostic variables".

Response: We will rephrase the sentence relying on reviewer's suggestion.

Modification: We rephrased the sentence which now reads:

"In this analysis, CLASS version 3.6 was used in offline mode and with a 30-min time step, ensuring an uninterrupted time series of the prognostic variables (Roy et al., 2013) hereby allowing the

inclusion of multiple soil layers and accounts for snow interception, snow thermal conductivity, and snow albedo, as described in Bartlett et al. (2006)."

17) Line 152: Change "weighting" to "weighing".

Response: Thanks, we will rectify this.

Modification: We have rectified this, thanks.

18) Line 152-3: I would acknowledge that there can be significant differences in snowfall for some snowfall events over a 4 km distance. It is probably not a consistent spatial bias, however, so the methodology is acceptable as long as there is such an acknowledgement.

Response: We agree with the reviewer and will add some sentences to acknowledge the bias present in the precipitation input.

Modification: We have added supporting reference and additional sentence to acknowledge the bias present in the precipitation input. Please refer to line 191 – 196.

19) Figure 2: Could site A1 be a brighter blue to aid in seeing it as distinct from site A4? The dots are otherwise fine as long as none are completely hidden. If any are completely hidden, perhaps some can have no fill or have the sizes made smaller.

Response: Thank you for relevant suggestion, we will change the color of the plot throughout the manuscript.

Modification: Thank you for this suggestion, we have now changed the color of plot for site Sap1 throughout the manuscript.

20) Figure 11: I am not sure what the light blue shading in the caption is referring to.

Response: Thanks, we will remove the light blue shading.

Modification: We have removed the light blue shading, thanks. Also, now Figure 11 is Figure 13.

Reference

Alves, M., Nadeau, D. F., Music, B., Anctil, F. and Parajuli, A.: On the performance of the Canadian Land Surface Scheme driven by the ERA5 reanalysis over the Canadian boreal forest, J. Hydrometeorol., 21(6), 1383–1404, doi:10.1175/jhm-d-19-0172.1, 2020.

Brun, E., Martin, E., Simon, V., Gendre, C. and Coleou, C.: An energy and mass model of snow cover suitable for operational avalanche forecasting, J. Glaciol., 35(121), 333–342, doi:10.1017/S0022143000009254, 1989.

DeWalle, D. R. and Rango, A.: Principles of snow hydrology, 1 st ed., Cambridge University Press, New York., 2008.

Hedstrom, N. R. and Pomeroy, J. W.: Measurements and modelling of snow interception in the boreal forest, Hydrol. Process., 12(10–11), 1611–1625, doi: 10.1002/(SICI)1099-1085(199808/09)12:10/11<1611::AID-HYP684>3.0.CO;2-4, 1998.

Jennings, K. S., Kittel, T. G. F. and Molotch, N. P.: Observations and simulations of the seasonal evolution of snowpack cold content and its relation to snowmelt and the snowpack energy budget, Cryosph., 12(5), 1595–1614, doi:10.5194/tc-12-1595-2018, 2018.

Molotch, N. P., Brooks, P. D., Burns, S. P., Litvak, M., Monson, R. K., McConnell, J. R. and Musselman, K. N.: Ecohydrological controls on snowmelt partitioning in mixed-conifer sub-alpine forests, Ecohydrology, 2, 129–142, doi:10.1002/eco.48, 2009.

Parajuli, A., Nadeau, D. F., Anctil, F., Parent, A.-C., Bouchard, B., Girard, M. and Jutras, S.: Exploring the spatiotemporal variability of the snow water equivalent in a small boreal forest catchment through observation and modelling, Hydrol. Process., 34(11), 2628–2644, doi:10.1002/hyp.13756, 2020b.

Russell, M., Eitel, J. U. H., Maguire, A. J. and Link, T. E.: Toward a novel laser-based approach for estimating snow interception, Remote Sens., 12(7), 1–11, doi:10.3390/rs12071146, 2020.

Seligman, Z. M., Harper, J. T. and Maneta, M. P.: Changes to snowpack energy state from spring storm events, Columbia River headwaters, Montana, J. Hydrometeorol., 15(1), 159–170, doi:10.1175/JHM-D-12-078.1, 2014.

**Reviewer 2**

The authors detail the evolution of snowpack cold content at 4 sites with differing vegetation cover properties in Quebec, Canada. They do this using a combination of field observations and model output over one field season at their research site. They detail differences in cold content across the sites and present extensive model validation plots and figures. There is a sufficient amount of in-depth analysis and it's clear the authors have devoted a great deal of time and care to the plots, tables, and manuscript. However, there are two major shortcomings/revisions that I feel need improvement before acceptance should be considered:

Dear Dr. Keith Jennings,

Thank you for the kind words and constructive comments. We are very confident that they will help us to build a new version with a much-improved interpretation of the results and a more straightforward take-home message.

1) Given the large number of multi-layer models available, why use a single-layer model and then attempt to split the layers?

Response: First, it is important to recall why we need to use a snow model. As shown in equation 1 of the paper, the cold content of the snowpack is computed from snow temperature, snow density,

and snow depth. During our snow pit surveys, we measured these variables jointly and calculated the associated *CC*. These results are presented in section 3 of the article.

However, several phenomena can occur between the snow pits, taken about every week. Thus, it is interesting to explore *CC* at a finer temporal scale. Thanks to our snow profiling stations, we dispose of 30-min measurements of snow temperature taken every 10-cm in the vertical direction, as well as snow depth. However, we do not have continuous measurements of density, and it is here that we decide to use a model.

Before using a model, it is essential to know its strengths and weaknesses. We chose CLASS because it was evaluated at our site for its ability to reproduce the observed snow heights (see Alves et al., 2020). The model showed its ability to reproduce the longevity of the snowpack but tended to underestimate the SWE for the reasons discussed in Reviewer 1's comment 1.

There are, of course, several other snow models available, some of which are multi-layered, but before resorting to them, we would have had to do a complete validation at our site, which we felt was beyond the scope of the study.

Since CLASS is a single-layer model, we employ a method to generate multiple density layers, much like Roy et al. did in 2013 to estimate the SSA. Note our process of densifying each layer taken from Shrestha et al. (2010) is similar to what is implemented in the snowpack model.

We, therefore, plan to clarify the reason for our use of CLASS in the new version of the paper by synthesizing the above points.

Modification: Please refer to line 401 – 407, where we have discussed the limitation of bulk estimate, uncertainty associated with both single and multi-layer model and why we have used CLASS in our present research.

2) It's particularly unclear to me how the Andreadis method is applied.

Response: In agreement with the other remarks along these lines, we will prepare the schematic diagram.

Modification: We propose to add Figure R5, which will clarify the methodology used to obtain the reconstructed *CC* time series. It shows where the application of Andreadis (2009) method fits into the process.

[Figure]

Figure R5: Schematic diagram showcasing the methodology adopted to conduct this study.

3) The results make it seem like the authors only track 3 layers, but then subsequent sections break out layers in 10 cm increments. Do the layers stay consistent or are they combined and divided as would be done in a multi-layer model?

Response: Indeed, we performed the 10-cm vertical layer scheme for our analysis and then divided them into 3 layers only to visualize and understand the difference of snow density across the snowpack. We believe, the newly prepared schematic diagram will clarify the raised concern.

Modification: Several modifications is applied to enhance the clarity of the methodology adopted in this study. Here, Figure R5 is now prepared to showcase the overall methodology adopted in this study. Also, in the result and discussion section we have exclusively described the usage of 3-layer scheme (Please refer to line 295 – 296 and 416 - 418).

4) The methods text says simulated SWE from CLASS is used, but then the equations are all for density.

Response: Thank you for your relevant suggestion. In our previous version, *Wns* and *Ws* (derived from CLASS) presented in Equation 6 represented the amount of new and old snow water equivalent, respectively.

Modification: To make this more explicit, we have changed the variable names to *SWEns* and *SWEs*, for new and old snow water equivalent, respectively. Please refer to Equation 7

5) Do the final simulated CC series use the thermocouple data or snowpack temperature output from CLASS? If the latter, how is temperature reallocated?

Response: We used thermocouple data, as shown in the new Figure R5.

Modification: Please refer to newly prepared Figure 2 which showcase the schematic diagram of methodology adopted in this study.

6) I think a schematic showing the complete workflow for the reconstructed CC time series would benefit the readers, in addition to providing much more detail in the methods text.

Response: This is a great idea, thanks and see previous comments 5.

7) Scientific contribution. I'm left wondering what the major novel contribution of the paper is, which means it needs further work to be accepted in The Cryosphere.

Response: The main original aspect of the work is as follows:

**Research topic**

As detailed in Jennings et al. (2018), the lack of direct observation still hinders the understanding of snowpack cold content. To the best of our knowledge, only a few studies have investigated this state variable. In the past, some observational studies (Seligman et al., 2014; Jennings et al., 2018) and snowmelt models have (Jost et al., 2012; Valery et al., 2014) either detailed or inserted snowpack cold content. However, only few of them attempted to understand this state variable at forest (e.g. Jost et al.,2012; Jennings et al., 2018). But none of them compared the variability of *CC* across different stand structures. In this study, we attempted to understand the *CC* variability across four contrasting sites with canopy. We believe, such comparison is first of its kind particularly for cold content studies.

**Methodology**

Aided by snow profiling station and labor-intensive field measurement, our study used a time series of snow temperature profiles (10-cm apart) and combined with an empirical formulation to produce a multi-layer cold content time series across four different forest sites. The manual weekly observation of multi-layer snow density, snowpack temperature, and snow depth enabled us to understand the behavior of *CC* across our forest sites. The use of 30 minutes 10-cm snowpack temperature profiles combined with snow depth and empirical snow density estimate provided us with a multi-layer time series of *CC*. We believe using an automated snowpack temperature profile at four sites to derive *CC* is the first of its kind as well.

**Take-home message**

Our study aimed to understand the *CC* behavior across four different forest stands. Based on our findings, we were able to relate the role of vegetation on the accumulation and ablation of snow and its role on *CC*.  Furthermore, we have presented the effect of short-lived events, such as the effect of cold air pooling on *CC*. Here site A3 exhibited a unique thermal regime which ultimately resulted in

the difference of *CC* at this site compared to others. We also would like to thank the reviewer for raising pertinent issues related to the inclusion of the rain-on-snow event to bolster our discussion. Indeed, the rain-on-snow event resulted in *CC* removal and led to almost uniform *CC* distribution across our sites (regardless of differences in vegetation and vegetation altered energy exchanges).

As presented in Jennings et al. (2018), the authors detailed the relationship between cumulative temperature and precipitation with cold content development. However, due to snow interception, snow accumulation and melt are more variable in forest than any other locations (Parajuli et al.,2020). In a forest, the intercepted snow may immediately unload or stay there (densify) for several days to months and unload (Dewalle and Rango, 2008). Thus, this study attempted to relate pertinent variables such as snow depth, air and snowpack temperature, and snow density. We believe this is also a novel aspect of our study.

8) Overall, my feeling is the authors can devote less results text to model validation and output (particularly the analysis of the 3-layer scheme as I note in my line-by-line comments below) and spend more time on process-based analysis. For example, what happens during the rain-on-snow events from an energy balance perspective? Why are there differences in cold content development across the sites? Is it the effect of canopy cover on snow accumulation and/or the snowpack energy balance? Given the field observations and modeling, I feel there is a lot more the authors could unpack that give us, the readers, deeper information on the processes governing cold content development and removal at these particular sites.

Response: We agree with the reviewer and add several sentences to describe different energy budget components and other process-based analyses in our discussion. This includes the role of the canopy on *CC* variability, the effect of rain-on-snow events on *CC* distribution, the importance of solar radiation and its role on *CC* variability, and the importance of snow interception on snow densification in forest and its role on *CC*. We will also compare different short-lived events from an energy balance perspective.

Modification: We applied several modifications to the manuscript by adding several sentences in the discussion section. This includes:

   a) Vegetation role: Please refer to line 442– 486 for more detail on role of vegetation snow interception and resulting variability in snow densification process.
   b) Effect of wind speed and solar radiation cumulation: Please refer to line 447 – 454 for more detail on effect of wind speed and solar radiation difference on snow accumulation difference and its role on *CC*.
   c) Rain-on-snow event: Please refer to line 479 – 486 for effect of rain-on-snow event on snowpack. For understanding the site-wise difference induced by rain-on-snow event please refer to Figure R3 (reviewer 1 response) or Figure 11 (manuscript).
   d) Cold air pooling:  Please refer to Figure R3 or Figure 11 for more detail on cold-air pooling with energy balance perspective.  Also please refer to line 465 – 475.

**Line-by-line comments:**

9) Line 9: Surface melt can begin before snowpack cold content = 0.

Response: We will remove the sentence, thanks.

Modification: We have removed the sentence.

10) Line 37: The snowpack energy balance predates USACE (1956). For example, Angström's work from the 1910s on the radiation budget and Western Snow Conference papers from the 1940s published in Transactions of the American Geophysical Union. I'm sure there's plenty more going back in time.

Response: Thanks for bringing this up, we will update the modify the sentence.

Modification: Please refer to line 38 – 40.

11) Line 37: DHSVM was developed with a single-layer snowpack, but now uses a two-layer formulation (Wigmosta et al., 2002): Wigmosta, M. S., Nijssen, B., Storck, P., & Lettenmaier, D. P. (2002). The distributed hydrology soil vegetation model. Mathematical models of small watershed hydrology and applications, 7-42.

Response: We agree with the reviewer. Thanks for your concern we will update it accordingly.

Modification: Please refer to line 42 where we added Wigmosta et al, 2002 as an additional reference.

12) Line 46: $\rho_w$ is not in the equation

Response: We will rectify the error, thanks.

Modification: We have placed the water density in appropriate location (Please refer to line 227)

13) Line 55: Change snow surveys to snow pits.

Response: We will change this, thanks.

Modification: Please refer to line 59.

14) Line 55: Reconsider "tedious and demanding." I'd probably say time-consuming (I find digging snow pits to be quite enjoyable, not tedious).

Response: We will rephrase the sentence, thanks.

Modification: Please refer to line 59.

15) Line 56: You can note example datasets here. E.g. the Williams and Morse Niwot LTER data: Williams, M., J. Morse, and Niwot Ridge LTER. 2020. Snow cover profile data for Niwot Ridge and Green Lakes Valley, 1993 - ongoing. ver 15. Environmental Data Initiative. https://doi.org/10.6073/pasta/a5fca9d02a4a6a0744cc0d0ffccacd09 (Accessed 2021-05-06).

Response: We will cite the reference you have provided, thanks.

Modification: Please refer to line 60.

16) Lines 56–59: This reads very similarly to the opening sentence of paragraph 2 in Jennings et al. (2018). Please note as such.

Response: We will reword the text to avoid too much similarity to Jennings et al. (2018) and add a reference to this paper.

Modification: Please refer to line 61 where we have added the reference.

17) Line 59: Please replace all uses of resort/resorted with a more appropriate word (e.g., used, leveraged, utilized, etc.). Resort typically has a negative connotation (i.e., there was nothing else we could use so we had to use this.).

Response: We will replace the word throughout the manuscript, thanks.

Modification: Similar to reviewer 1 comments we have avoided using word "resorted" where necessary.

18) Lines 62–64: This was noted in Jennings et al. (2018) (please add text as such). Lines 65–66: This is again very similar to the language in the Jennings et al. (2018) manuscript. Please be careful with paraphrasing and providing citations for paraphrased work.

Response: We will improve the sentence.

Modification: Please refer to line 73 – 77 for more details.

19) Line 68: Jennings et al. (2018) includes a forested site (the subalpine) and compares it to a higher, open site (the alpine).

Response: We agree with the author and would modify the sentence.

Modification: The sentence now read:

"Little to no previous research has focused on comparing the *CC* behaviour across variable stand structures within forest environments ".

20) Line 69: Remove "obviously"

Response: We will remove it, thanks.

Modification: Please refer to line 81.

21) Lines 78–79: "model articulated around the energy balance" > rephrase for clarity.

Response: We will rephrase the sentence, thanks.

Modification: The sentence now reads:

"One such example is the Canadian LAnd Surface Scheme (CLASS), which relies on a single-layer snow model based-on the energy balance ".

22) Line 84: "even models such as this are not free of biases" > rephrase because all models have some form of bias. You could highlight the previously cited snow model comparison literature.

Response: We will add a supporting sentence for our statement.

Modification: the sentence now reads:

"For instance, Raleigh et al. (2016) tested three physically based snow models (Utah Energy Balance (UEB), Distributed Hydrology Soil Vegetation Model (DHSVM) and Snow Thermal Model SNTHERM) and reported biases in longwave radiation estimation ranging from −12 to +18 W m$^{-2}$."

23) Line 87: What is "acceptable" in this context? Most readers, myself included, will not have a good intuition of what large and small SSA values are.

Response: Thank you for your concern. We will add additional details.

Modification: The sentence now reads:

"For instance, Roy et al. (2013) disaggregated CLASS-derived snow water equivalents into multilayer values at each time step, for the purpose of estimating the specific surface area (SSA) of a snowpack. In their study, the authors reported a specific surface area ranging from 33.1 to 155.8 m2 kg$^{-1}$, while attaining an acceptable root mean square error (RMSE) of 8.0 m2 kg$^{-1}$ in CLASS-derived SSA for individual layers."

24) Line 89: Remove "obvious" and rephrase sentence. I think you're referring to cold content modeling specifically, but there are many forest snow model studies.

Response: Based on the reviewer suggestion, we will rephrase the sentence.

Modification: The sentence now reads:

"In view of the lack of observational studies (particularly on *CC*) that are required to support model development in forest environments, detailed analyses of multilayer in situ snowpack CC are necessary".

25) Line 105 (figure 5 caption and throughout): Consider changing the names A1 through A4 to more meaningful terms or abbreviations. There are three forest cover classes (sapling, juvenile and mature). You could use some variation thereof and specify which mature forest site is denser.

Response: We will change the name to "Sap1", "Juv1", "Mat1" and "Mat2" instead of A1 through A4.

Modification: We have changed the name of the site throughout the manuscript, thanks.

26) Lines 117–124: This could use greater explanation. What instruments were used, what was the vertical frequency of density and temperature sampling, etc? How were the pit values used to impute the missing data?

Response: The snow pit data (snowpack density, temperature, and depth) are exclusively used for the observational study (sections 3.1, 3.2, and 4.1). For model validation, we used the snow density from snow pit surveys. However, the snow profiling station provided the 30-minute 10-cm snowpack temperatures.

Modification: Please refer to newly prepared Figure 2 for more details.

27) Lines 128–130: "A simple approach would be to interpolate the density values extracted from snowpits, but this would be incomplete and error-prone given their limited number and absence early in the season." This seems overly subjective, especially considering the low $r^2$ values for the hybrid approach.

Response: Our site has the particularity of being very snowy, and it is not uncommon for episodes delivering >10 cm of fresh snow to occur (see Figure 8 from the paper). In this sense, a simple linear interpolation of density values appears to be a hazardous exercise. Also, as stated in the cited sentence, if we were to do this, we would have no value during the early season and very few towards the end of the melting period. This will be emphasized in the next version of the paper.

Modification: Please refer to line 146 – 149.

28) Lines 136–137: Further explanation needed. For example, what are the prognostic variables?

Response: We thank reviewer for raising this issue. We will add the prognostic variables.

Modification: The sentence now reads:

"In this analysis, CLASS version 3.6 was used in offline mode and with a 30-min time step, ensuring an uninterrupted time series of the prognostic variables (Roy et al., 2013) hereby allowing the inclusion of multiple soil layers and accounting for snow interception, snow thermal conductivity, and snow albedo, as described in Bartlett et al. (2006)."

29) Line 147: Change Table 2 to Table 1

Response: Thank you for your suggestion. We will change this.

Modification: Please refer to line 175.

30) Table 2: It doesn't seem like all sites should be marked as having PPT measurements when the text says it was measured at one location 4km away?

Response: We agree with the reviewer and will modify the table accordingly, thanks.

Modification: We have now added the separate subdivision to our Table 2 by adding the ECCC site from where precipitation data was extracted.

31) Line 165 (Section 2.2.3): Please see my major revision comment #1. This section needs significant improvement in terms of clarity and specificity.

Response: Thank you for raising this issue. We have prepared a schematic diagram (Fig. R4) and will take great caution at explaining every step of the approach.

Modification: We have now added additional information in this section 2.2.3 where a schematic diagram is prepared, and some problematic sentences are modified. Please refer to this section for more detail.

32) Lines 169–171: Text says minimum, but equation uses maximum.

Response: Indeed, the text says minimum as we took the minimum value of snow density derived by snow pit surveys which is 76 kg m$^{-3}$. However, the use of Equation 3 sometimes resulted in snow density estimates below 76 kg m$^{-3}$. Thus, we used a maximum value as in Brun et al. (1989).

33) Lines 206–207: Are these differences computed within the snowpack at each site per time or computed across all sites?

Response: Here, we presented the minimum and maximum cold content from all the sites for individual layer. We will clarify this in the next version.

Modification: Please refer to line 249.

34) Line 213 (and throughout text): Please change amplitude to magnitude.

Response: We will change them throughout the manuscript, thanks.

Modification: The word is now changed throughout the manuscript, thanks.

35) Figure 3: I like the amount of info conveyed by this figure, but find the color scale to be confusing. Because it's representing a single, non-divergent variable, you would be better suited by using the shade of single color. Also, please add "Layer cold content" to the scale bar.

Response: The reviewer raised a good point. We have prepared a new version of the plot with shades of a single colour (Fig. R5). We find it more difficult to make a quantitative interpretation of

the figure and would prefer to keep the original version.

[Figure]

Figure R5: Example of multi-layer cold content observation using the shades of the purple colour.

36) Line 222: Is this the maximum difference or difference in maximum CC? Table 3 shows a larger difference between A1 and A4.

Response: Indeed, here we are referring to the mean total *CC*. As such, we believe this statement is correct.

37) Table 3: The minimum appears to occur in late March (early April?) for site A2 as seen in Fig 4.

Response: True. We will correct this mistake, thanks.

Modification: We have now changed the date of occurrence of peak CC as 2018-03-20 for site Juv1, thanks.

38) Line 232: Please use less subjective terms (for example, "CLASS produces low mean biases in...").

Response: Thanks, we will modify the sentence.

Modification: Please see lines 280 – 283 for more details.

39) Line 233: Somewhat interestingly, the negative SWE bias would indicate the model has a cold bias in snowpack temperature in order to get CC so accurately. Please include snowpack temperature in the validation.

Response: See our answer to comment 1 from Reviewer 1.

40) Line 239: It appears you're producing a 3-layer scheme, but this is not specified in the methods section. Please add that along with information on how the layers are defined/combined/separated (see major revision comment #1).

Response: See our answer to your comment 3.

41) Lines 240–242: You need to clarify when you're using snow temperature from the thermocouple arrays versus from the snow pit or CLASS. It's not clear here or in the methods section. I think it would be worth adding some material to the methods and again to the results so the readers are certain when observed versus simulated values are being used.

Response: The new figure (Fig. R5) will avoid any confusion in this regard.

42) Line 249: This statement seems incorrect. If you're using the snow temperature from the thermocouple, then most of the error is coming from the density estimates and layering scheme.

Response: True, most of the error is coming from density estimate. Thus, we have portrayed Figure 11 (now figure 13), to identify the probable reason for uncertainty in snow density estimate.

43) Figure 6 and preceding text: I'm leaning towards getting rid of the arbitrary 3-layer scheme and only validating/describing the 10 cm layer results shown below. It's unclear what information/utility the 3-layer scheme provides given that most of the findings are based on the observations and the 10 cm layer discretization.

Response: It is a way to visualize a large amount of data. To reproduce the same figure but for 10-cm slices would be very difficult to analyze in our opinion.

44) Figure 7: Please change to a table. The color scale provides the same information as the numbers, but the numbers on their own are easier to interpret.

Response: We prefer to keep the plot as it is.  A similar figure is presented in Parajuli et al. (2020a) and we believe there are several advantages to it. For instance, the colour box immediately notifies the reader about the existing difference in our error metric. Also, the text will provide a minor detail present in this plot. We believe, both text and color scale are complimenting each other.

45) Lines 264–265: Were the rain-on-snow events missing because no snow pits were dug at those times or the snow pits suggested different changes to cold content than the simulations? Either way, this needs to be clarified.

Response: Indeed, no snow pits were dug during rain-on-snow events. Please refer to Line 132 for more details.

46) Figure 8: Same comment as figure 3. Consider changing the color ramp and adding "Layer cold content" to the scale bar.

Response: Please refer to comment 35, thanks.

47) Lines 273–274: This assertion is presented without supporting evidence. In line with my major revision comment #2, this would be an ideal place to test some process-based hypotheses.

Response: We disagree with the reviewer. Parajuli et al. (2020b) reported the effect of stand structure on snow accumulation and ablation in the present research site, which is acknowledged in the discussion section. Here, as presented in Table 1, the stand structure is not uniform. Also, differences in snow accumulation and ablation are shown in Figure 9b, influenced mainly by vegetation. I believe this assertion is not presented without supporting evidence. However, we will add some more details to justify our statement.

Modification: Please see line 329 – 330 where we have now added the Figure 9 as a reference where snow accumulation and melt pattern are different at our sites of interest and Table 1 where difference of vegetation is being displayed.

48) Figure 9: Why is simulated cold content plotted with observed snow depth? It seems like observed snow depth should be plotted with earlier observational figures. Also, keep the rain-on-snow shading consistent with previous figure.

Response: Here, our simulation is largely built from observations. We believe the new figure (Fig. R5) helps clarify this. We agree with the reviewer for the rain-on-snow shadings and will modify this accordingly.

Modification: We have now modified the rain-on-snow shading in our Figure 10.

49) Lines 281–285: These results need more unpacking and their associated methods need to be moved to the methods section. You should also be careful with "positive" and "negative" correlations here. Most cold content values have been discussed in terms of their magnitude, which may lead readers to think cold content declines (i.e. approaches 0) when air temperature decreases. Additionally, simulated and observed values need to be noted explicitly in the text along with the period of comparison (is CC correlated with 30-min air temperature, daily air temperature, or average air temperature to date?).

Response: We will be more transparent in this section. We will move all material associated with the methodology to the appropriate section. Also, the negative correlation is due to negative values of *CC* and positive values of snow depth. We will modify the text accordingly.

Modification: We have now modified the sentence (Please refer to line 351 - 353) and added the probable reason for negative correlation in our discussion (Please refer to line 491 - 493), thanks.

50) Lines 286–291: Similar to my above comment, this needs much further explanation. This section could become an important part of the paper (and its novel contribution to the field) if you can further evaluate forest cover differences and their quantitative effect on cold content evolution. For example, you could include an assessment of snowpack energy balance differences and/or changes in snow accumulation as caused by forest cover in both observations and the model.

Response: We disagree with the reviewer and believe that we have evaluated the forest cover difference and their quantitative role on *CC* evolution. Figure 3 to Figure 10 detail the spatiotemporal

variability of cold content with respect to the vegetation. As illustrated in Figure 3 and Figure 4, based on observations, the detailed multi-layer cold content time series is presented for respective sites with diverse stand structures.  In Figure 9, we sought the relationship between the snow depth (accumulation and ablation of snow) and its role on *CC*. Figure 10 attempted to establish the relationship between different pertinent vegetation inputs with *CC*. In this plot, we performed such an assessment detailing differences in the vegetation. In the result section, we have described the mean, peak, and even the variability of *CC* across sites. Nonetheless, we will improve and add more relevant information as suggested by both reviewers. We agree with the reviewer, and we will now present the *CC* variability and short-lived events with a snowpack energy budget perspective. Please refer to comment 8.

Modification: Please refer to comment 8.

51) Figure 10: Please change to table (same comment as figure 7).

Response: Please refer to comment 44.

52) Line 298–299: Figure 11 does not show cold content.

Response: Indeed, the cold content time-series for site Sap1 was plotted before (Please refer to Figure 3)

53) Figure 11: Need to clarify if these are simulated or observed values. There is no light-blue shading in the plots. Also, the color ramp should not be divergent as the values they represent are not. Consider using gradation of single color.

Response: We agree with the reviewer and will modify accordingly, thanks. For colour ramp please see the comment above.

Modification: We have modified the caption which now reads:

"Observed 10-cm snowpack temperature (top) and density (bottom) at site Sap1. "

54) Lines 304–309: Figure 4 does not show mass, only cold content. It might be worth further evaluating SWE, depth, and cold content over time in the results sections. Also note that the average winter temperatures in Jennings et al. (2018) were ~4°C cooler at the alpine site (colder frozen mass and more of it led to greater CC in the alpine).

Response: True, Figure 4 is based on weekly measurements, and we don't have complete series of SWE and depth. However, as presented in Figure 9, we have plotted *CC* with the snow depth as the snow-profiling stations provided continuous time series of snow depth.

Response: Thank you for providing the detail description of Jennings et al. (2018), we will add this information.

Modification: We have added a sentence to showcase the difference in temperature between alpine and subalpine site (Please refer to line 384).

55) Lines 318–350: ) Lines 318–350: I like this section, but I feel like the paper would have a greater impact if there was a greater reliance on results versus discussion when comparing the sites (please see my comment on lines 286–291). For example, you discuss cold air pooling here, but don't provide data. Why not add this to the results section with data from the air temperature sensors at the different sites? If these data don't support the hypothesis, then it can be removed. Data from Jennings et al. indicated that the energy balance was typically positive when snow was not actively accumulating. However, there were exceptions at night as a result of radiative cooling from the snowpack. You could provide a comparison from your sites here by providing energy balance output from CLASS in the results.

Response: We agree with the reviewer and will modify the cold air pooling discussion. As pointed out by reviewer 1, we will add the discussion relating to the low wind speed (stable atmospheric condition) and cold air pooling mechanism, referring to Figure R2. Also, we will compare the energy balance output for the period with low wind speeds and rain-on-snow events.

Modification: In our recent version of manuscript, we have modified Figure 3 which now details the low wind speed period coinciding with cooler temperature at site Mat1. We have now prepared new figure (Fig. 11) to showcase the energy balance perspective with different period such as cold air pooling, rain-on-snow event, spring and winter period. In our discussion the several information relating the different periods to bolster our discussion as well.

56) Figure 12: This is an unfair comparison. The density of freshly fallen snow is not comparable to the density of the top 10 cm of a snowpack. There's no fresh snow I know of that falls at 400 kg m$^{-3}$.

Response: Sorry for misunderstanding. However, the plot displays the superficial snow density not the fresh one.  Please refer to Figure R4

57Lines 362–366: Please see comment above. Consider removing plot and text unless the analysis is changed to provide a more important validation of new snow.

Response: Please refer to above comment 56.

58) Line 397: Please add link in revised manuscript.

Response: At this point, we are working on a data manuscript and will share our data once our paper undergoes discussion. Also, the dataset will be available from authors on a reasonable request immediately. Thanks.

Reference

Alves, M., Nadeau, D. F., Music, B., Anctil, F. and Parajuli, A.: On the performance of the Canadian Land Surface Scheme driven by the ERA5 reanalysis over the Canadian boreal forest, J. Hydrometeorol., 21(6), 1383–1404, doi:10.1175/jhm-d-19-0172.1, 2020.

Andreadis, K. M., Storck, P. and Lettenmaier, D. P.: Modeling snow accumulation and ablation processes in forested environments, Water Resour. Res., 45(5), 1–13, doi:10.1029/2008WR007042, 2009.

Bartlett, P. A., MacKay, M. D. and Verseghy, D. L.: Modified snow algorithms in the Canadian land surface scheme: Model runs and sensitivity analysis at three boreal forest stands, Atmosphere-Ocean, 44(3), 207–222, doi:10.3137/ao.440301, 2006.

Brun, E., Martin, E., Simon, V., Gendre, C. and Coleou, C.: An energy and mass model of snow cover suitable for operational avalanche forecasting, J. Glaciol., 35(121), 333–342, doi:10.1017/S0022143000009254, 1989.

DeWalle, D. R. and Rango, A.: Principles of snow hydrology, 1 st ed., Cambridge University Press, New York., 2008.

Jennings, K. S., Kittel, T. G. F. and Molotch, N. P.: Observations and simulations of the seasonal evolution of snowpack cold content and its relation to snowmelt and the snowpack energy budget, Cryosph., 12(5), 1595–1614, doi:10.5194/tc-12-1595-2018, 2018.

Jost, G., Moore, R. D., Smith, R. and Gluns, D. R.: Distributed temperature-index snowmelt modelling for forested catchments, J. Hydrol., 420–421, 87–101, doi:10.1016/j.jhydrol.2011.11.045, 2012.

Parajuli, A., Nadeau, D. F., Anctil, F., Schilling, O. S. and Jutras, S.: Does data availability constrain temperature-index snow model? A case study in the humid boreal forest, Water, 12(8), 1–22, doi:10.3390/w12082284, 2020a.

Parajuli, A., Nadeau, D. F., Anctil, F., Parent, A.-C., Bouchard, B., Girard, M. and Jutras, S.: Exploring the spatiotemporal variability of the snow water equivalent in a small boreal forest catchment through observation and modelling, Hydrol. Process., 34(11), 2628–2644, doi:10.1002/hyp.13756, 2020b.

Raleigh, M. S., Livneh, B., Lapo, K. and Lundquist, J. D.: How does availability of meteorological forcing data impact physically based snowpack simulations?, J. Hydrometeorol., 17(1), 99–120, doi:10.1175/JHM-D-14-0235.1, 2016.

Roy, A., Royer, A., Montpetit, B., Bartlett, P. A. and Langlois, A.: Snow specific surface area simulation using the one-layer snow model in the Canadian LAnd Surface Scheme (CLASS), Cryosph., 7(3), 961–975, doi:10.5194/tc-7-961-2013, 2013.

Seligman, Z. M., Harper, J. T. and Maneta, M. P.: Changes to snowpack energy state from spring storm events, Columbia River headwaters, Montana, J. Hydrometeorol., 15(1), 159–170, doi:10.1175/JHM-D-12-078.1, 2014.

Shrestha, M., Wang, L., Koike, T., Xue, Y. and Hirabayashi, Y.: Improving the snow physics of WEB-DHM and its point evaluation at the SnowMIP sites, Hydrol. Earth Syst. Sci., 14(12), 2577–2594, doi:10.5194/hess-14-2577-2010, 2010.

Valéry, A., Andréassian, V. and Perrin, C.: 'As simple as possible but not simpler': What is useful in a temperature-based snow-accounting routine? Part 2 – Sensitivity analysis of the Cemaneige snow accounting routine on 380 catchments, J. Hydrol., 517, 1176–1187, doi:10.1016/j.jhydrol.2014.04.058, 2014.

**Other minor correction**

Change of title: Multilayer observation and estimation of the snowpack cold content in a humid boreal coniferous forest of eastern Canada

Line 5: Change of email ID.

Line 27: Change from R2 value of 0.90 to 0.93 percent bias from −2.0% to −3.3% (Minor error of some inputs detected in the code). Not a major change.

Line 152 – 154: Added new sentence to describe the schematic diagram.

Equation 3: Changed 273.16 to 273.15 for uniformity

Line 202: Except for some plots the air temperature was in Kelvin so we removed the incorrect sentence.

Line 294 – 295: We found the small error as mentioned above in the Line 27 and rectified it. This affected the result presented in this line and two figures i.e., Figure 7 and 8. This has been modified

Data availability: We have rephrased the sentence " The data that support the findings in this study are available upon request to the main author."

---

## Referee Report (RR1)

**Overview**
The authors present a revised manuscript detailing the seasonal evolution of cold content at four sites in the Boreal forest of Québec, Canada. This version is markedly improved, particularly the introduction and results/discussion. The methods are easier to follow, making the paper more accessible to a wider audience. I recommend publication after a couple minor revisions. These are shown in the list below and detailed more thoroughly in my line-by-line comments.

**Minor revisions**
1. Lines 313–325. Persistently negative net fluxes often translate into cold content magnitudes that are too high. Please check the numbers here and the allocation of energy losses in CLASS.
2. Lines 410–442: Please move novel results to results section and save this space for your discussion text of the energy balance differences.

**Line-by-line**
Line 11: Change "Estimating the cold content" to "Measuring cold content"
Line 16: Add "modeled" before time series
Lines 25–27: Please clarify that CLASS output is compared to bulk values and that the empirical scheme is compared to layered values (unless I misread)
Lines 36–37: Not important, but you can typically strengthen sentences by removing words (i.e., "improved decision-making in the area of water resources management" becomes "improved water resources decision-making")
Line 95: Please modify to note SSA is a property of snow grains
Line 99: Although I appreciate the shout-out, you can remove Jennings et al. (2018) as you've done novel, independent research
Line 140: Change "it was opted" to "we opted"
Line 145 (Figure 2): The schematic makes the process much easier to understand. The only suggestion I have is that directional arrows in a flowchart typically denote the creation of a new variable from a previous variable. In this context, arrows ps, HS, and Ts in the lefthand box would point to a new variable called cold content (not snowpit). Given these measurements are all from the snowpit, the lefthand box can be annotated to say weekly snowpit. I think the flowchart is probably fine as-is, but can be improved for clarity.
Line 247 (Figure 4): There's no light blue shading
Line 269: You can also cross-reference your methods subsection here
Line 275: Change "structured" to "summarized"
Line 290 (Figure 8): We'll agree to disagree here (if you change your mind, please remove the shading and make this a table as the shading provides no additional info past the numbers shown)
Line 306: Should Fig. 8 be 10?
Lines 313–325: I am suspicious about the persistent negative flux in the WA period. In general, cold content is a small proportion of the total energy required to warm, ripen, and melt a snowpack. Therefore, persistently negative $Q_m$ values over long periods are often improbable because they lead to impossibly low cold content values. For example, if we assume a WA

period of 90 days, which is a conservative assumption, the low $Q_m$ value of -6.6 W m$^{-2}$ at Sap1 would produce -51.3 MJ m$^{-2}$ of cold content (it's not clear whether cold content added by snowfall is included in $Q_m$ or if CLASS adds it separately). Perhaps the duration of rain-on-snow events with high positive energy fluxes is enough to counteract this effect, but I'm still skeptical. If we take the December 17$^{th}$ to January 18$^{th}$ period (consistent CC development), CC goes from ~-1 MJ m$^{-2}$ to ~-4 MJ m$^{-2}$ at Sap1. This corresponds to an average flux of just -1.1 W m$^{-2}$, which may or may not include the CC added by new snowfall. Perhaps the persistent negative values of $Q_m$ are "correct" but CLASS is using them to also cool the soil layer and not just the snowpack? No matter the cause, this discrepancy between the CC values and the negative fluxes needs further investigation and clarification. (One note: Subdaily net fluxes can be negative—e.g., nighttime cooling or cold, windy periods of high sublimation—but these would be balanced out over longer time scales so that the average net flux was near zero during the winter accumulation season.)

Line 327: Remove extra period

Line 328: Add clarification after "positive correlation" (i.e., "meaning cold content magnitude increased as air temperature and snowpack temperature decreased")

Line 341: The air temperature values are nearly identical. Please change

Line 343: Figure 13 does not show cold content

Lines 341–347: On closer reading, this paragraph and figure can be removed. The text is difficult to follow and it's unclear how the figure contributes to the discussion past the figures already displayed.

Lines 356–361: An alternative hypothesis is that a cold snap led to snowpack cooling. The more exposed site (Sap1) experienced greater cooling than the less exposed sites.

Lines 365–366:  Change "Based on our findings, CLASS successfully estimated snow density and *CC*. Although CLASS 365 reasonably predicted *SWE*, one cannot deny the fact that it underestimated observations" to "Based on our findings, CLASS successfully estimated snow density and *CC*, while underpredicting SWE."

Lines 367–371: Are these sentences suggesting class *overestimates* sublimation, leading to an *underestimation* of SWE? If so, please clarify

Lines 380–407: This section should be shortened significantly. I think it is sufficient to say the new snow density formulations led to persistent underprediction errors for the top layers, even when compaction is considered. This is likely a shortcoming of both the fresh density and compaction due to metamorphosis equations.

Lines 410–442: I really like the inclusion of the energy balance analysis, but these sentences contain novel results. Please keep the discussion parts here while moving the quantifications to an additional results subsection. (Also, please limit the use of "indeed" to start sentences.)

Line 417: Remove "at" from in front of "canopy"

Line 419: My take on this is that the 0.1 MJ m$^{-2}$ difference is negligible given the observation precisions and model biases.

Line 434: Please expand on this sentence. The canopy blocks incoming solar radiation (more CC) but also increases incoming longwave radiation (less CC) compared to an open site.

Line 437: Please cite peer-reviewed literature on energy fluxes during rain-on-snow (e.g., Marks et al., 1998): https://doi.org/10.1002/(SICI)1099-1085(199808/09)12:10/11%3C1569::AID-HYP682%3E3.0.CO;2-L

---

## Author Response (AR2)

Dear editor,
We would like to thank you for providing us with the opportunity to revise our paper. Also, we thank both reviewers for providing constructive feedback and helping us improve the overall quality of our manuscript. As detailed below, we have either modified the paper or provided proper justification to address the reviewers' concerns.

**Reviewer 1**

The authors have carefully considered the comments from myself and the other reviewer, and I am satisfied that they have sufficiently addressed my concerns and suggestions. I do have some minor corrections and suggestions that do not alter the substance of the paper.

Thank you for your kind words.

Suggested minor corrections and changes:

1) Line 65: This is confusing. The cold content could be very negative such that the snow does not melt, as happens for most of the winter. As such, how is a melt delay defined? I can understand reporting that when the cold content was estimated to be zero at 6:00 AM, melt was not detected for another 2.3 - 2.8 hours (perhaps until the radiation balance was no longer negative, or radiation plus sensible heat flux [i.e. the snowpack energy balance] was > 0).

Sorry for this confusion. We have removed this confusing sentence, thanks.

2) Line 66-67: The delaying effect of CC on melt in spring is less because, while air temperature must be at or above 0°C in all cases, in spring the radiation input is greater, which would lessen the delay because of increased energy available for melt.

Thank you for your suggestion, we have modified the sentence, which now reads: "This delaying effect of CC on melt is less marked in spring (Seligman et al., 2014), when excess radiative energy contributes more significantly to melt."

3) Line 68-69: Good point!

Thanks.

4) Line 116: Delete "of".

We deleted it, thank you.

5) Line 180: Change "weighting" to "weighing".

Done, thanks.

6) Line 240: Change "Excpet" to "Except".

We rectified this mistake, thank you.

7) Line 242 & Figure4: I wonder if the largest CC at Sap1 was missed in week 5. Perhaps note this possibility.

Thank you for pointing this out. We have added the following sentence: "Due to logistical constraints, we were unable to sample the Sap1 site on week 5, when all other sites appear to report one of the largest cold content amplitudes of the winter."

8) Table 3 & Figure 5: The smallest CC at Mat1 appears to be March 20 or on the same date as Sap1 and Mat2 based on Figure 5.

Thank you for pointing out this minor mistake. We have rectified it, please refer to Table 3.

9) Line 275: Do the authors mean Figure 8?

Indeed, thank you.

10) Line 279: Perhaps the density of the top layer is underestimated because of CLASS not accounting for the densification of snow prior to unloading, as well as possible drip from a solar-heated canopy that would freeze. A low density would result in a smaller heat capacity and bias CC towards smaller (less negative) values (although thermal conductivity would also be affected).

Good point! We have now added a sentence to discuss the effect of radiative heating of the canopy resulting in drip from intercepted snow and densification of the top snow layer in the discussion section: "In addition, the intercepted snow could be heated by radiation incident on the canopy. In doing so, local melting could be observed, resulting in dripping and an increase in the density of the top snow layers. It is possible that CLASS does not capture this phenomenon well."

11) Line 319: Do the authors mean "SM" rather than "SP"?

We have rectified this error, thank you.

12) Line 341: Change "unique" to "different".

Done, thanks.

13) Line 350: Do the authors mean "Jun1" rather than "Sap2"?

Indeed, we have corrected this error.

14) Line 376: Change "certain" to "a certain".

Done, thanks.

15) Line 381: Change "generated a reasonable" to "generated reasonable".

Done, thanks.

16) Line 382: Change "that are more prone to capturing short-lived" to "that are better able to capture short-lived".

Thank you for your suggestion, we have modified our sentence.

17) Line 403: Drip from a canopy that is heated by solar radiation is also a factor here.

Please see comment 10 above.

18) Line 409-410: Remove one instance of the word "indeed".

Done.

19) Line 413-14: The effect of high winds during cold air messes is as described. However, often under stable conditions radiative cooling is the dominant mechanism and wind would warm the snowpack during such times.

Indeed, this is why we provided a short discussion about sensible heat fluxes (lines 408 to 411).

20) Line 416-17: I am confused by "The absence of a well-defined at canopy also means greater incoming shortwave radiation."

We removed "at" from our sentence, thanks.

21) Line 420: Change "This is also the reason why there were 60% occurrence where magnitude of CC"to "This is also the reason why 60% of the time, the magnitude of CC".

We have changed this sentence accordingly, thanks.

22) Line 422: Change "For site Mat1, there were 32% occurrences where magnitude of CC were higher than at Sap1," to For Mat1, the magnitude of CC was higher than at Sap1 32% of the time".

Done.

23) Line 472-3: Since this was not absolute, I would either turn it around and state that "Areas with taller vegetation had the smallest snow accumulation and thus resulted in the smallest peaks in total CC" or state "The two sites with lower vegetation had greater snow accumulation and larger peaks in total CC relative to the sites with taller vegetation".

We have changed this sentence according to your suggestion, thanks.

24) It would be good if the data could be posted somewhere accessible.

Data are available upon request. We also plan to publish them in a data journal soon.

**Reviewer 2**
The authors present a revised manuscript detailing the seasonal evolution of cold content at four sites in the Boreal forest of Québec, Canada. This version is markedly improved, particularly the introduction and results/discussion. The methods are easier to follow, making the paper more accessible to a wider audience. I recommend publication after a couple minor revisions. These are shown in the list below and detailed more thoroughly in my line-by-line comments.

Thank you for your kind words.

Minor revisions
1. Lines 313–325. Persistently negative net fluxes often translate into cold content magnitudes that are too high. Please check the numbers here and the allocation of energy losses in CLASS.

We agree with reviewer that persistent negative net fluxes often translate into large magnitude of CC. It is also correct that our research site persistently experienced cooler temperature from late December 2017 to the end of January 2018, which might have negatively skewed the melt/refreeze flux. Afterwards, there were rain-on-snow events, cold spells, and episodes with milder air temperature. At our site, the snow stayed for 218 days consisting of 176 days of snow accumulation and 42 days of snow ablation. During the winter, sunshine duration ranges from 7 – 9 hours whereas it ranges from 13 – 15 hours in springtime. Based on your suggestion, we carefully checked CLASS simulation, and the values look ok. Nevertheless, we have to acknowledge the fact that several errors and bias may arise when simulating CLASS. One source of error (latent heat overestimation) is highlighted in our discussion section. Also, Alves *et al.* (2020) already pointed out some deficiencies of CLASS for the present research site.

2. Lines 410–442: Please move novel results to results section and save this space for your discussion text of the energy balance differences.

See our answer to your comment 26.

Line-by-line
3) Line 11: Change "Estimating the cold content" to "Measuring cold content"

We rectified this concern, thank you.

4) Line 16: Add "modeled" before time series

Thank you for your suggestion. It is added.

5) Lines 25–27: Please clarify that CLASS output is compared to bulk values and that the empirical scheme is compared to layered values (unless I misread)

Thank you for your suggestion. We have addressed your concern.

6) Lines 36–37: Not important, but you can typically strengthen sentences by removing words (i.e., "improved decision-making in the area of water resources management" becomes "improved water resources decision-making")

We modified the sentence, thanks.

7) Line 95: Please modify to note SSA is a property of snow grains

We addressed your concern, thanks.

8) Line 99: Although I appreciate the shout-out, you can remove Jennings et al. (2018) as you've done novel, independent research

We prefer to keep this sentence as it is, thanks.

9) Line 140: Change "it was opted" to "we opted"

We changed the sentence, thanks.

10) Line 145 (Figure 2): The schematic makes the process much easier to understand. The only suggestion I have is that directional arrows in a flowchart typically denote the creation of a new variable from a previous variable. In this context, arrows ps, HS, and Ts in the lefthand box would point to a new variable called cold content (not snowpit). Given these measurements are all from the snowpit, the lefthand box can be annotated to say weekly snowpit. I think the flowchart is probably fine as-is, but can be improved for clarity.

We have modified our schematic accordingly, thanks.

11) Line 247 (Figure 4): There's no light blue shading

We rectified our error, thanks.

12) Line 269: You can also cross-reference your methods subsection here

Based on your suggestion, we have added the cross-reference. Thanks.

13) Line 275: Change "structured" to "summarized"

Done, thanks.

14) Line 290 (Figure 8): We'll agree to disagree here (if you change your mind, please remove the shading and make this a table as the shading provides no additional info past the numbers shown)

We prefer the keep the plot as it is, thanks.

15) Line 306: Should Fig. 8 be 10?

True, we changed the mistake, thanks.

16) Lines 313–325: I am suspicious about the persistent negative flux in the WA period. In general, cold content is a small proportion of the total energy required to warm, ripen, and melt a snowpack. Therefore, persistently negative $Q_m$ values over long periods are often improbable because they lead to impossibly low cold content values. For example, if we assume a WA period of 90 days, which is a conservative assumption, the low $Q_m$ value of -6.6 W m-2 at Sap1 would produce -51.3 MJ m-2 of cold content (it's not clear whether cold content added by snowfall is included in $Q_m$ or if CLASS adds it separately). Perhaps the duration of rain-on-snow events with high positive energy fluxes is enough to counteract this effect, but I'm still skeptical. If we take the December 17th to January 18th period (consistent CC development), CC goes from ~-1 MJ m-2 to ~-4 MJ m-2 at Sap1. This

corresponds to an average flux of just -1.1 W m-2, which may or may not include the CC added by new snowfall. Perhaps the persistent negative values of Qm are "correct" but CLASS is using them to also cool the soil layer and not just the snowpack? No matter the cause, this discrepancy between the CC values and the negative fluxes needs further investigation and clarification. (One note: Subdaily net fluxes can be negative—e.g., nighttime cooling or cold, windy periods of high sublimation—but these would be balanced out over longer time scales so that the average net flux was near zero during the winter accumulation season.)

Please see the response to comment 1, thanks.

17) Line 327: Remove extra period

We removed it, thanks.

18) Line 328: Add clarification after "positive correlation" (i.e., "meaning cold content magnitude increased as air temperature and snowpack temperature decreased")

We have added this clarification, thanks.

19) Line 341: The air temperature values are nearly identical. Please change

This paragraph has been removed as you suggested in comment 21.

20) Line 343: Figure 13 does not show cold content

This figure has been removed as you suggested in comment 21.

21) Lines 341–347: On closer reading, this paragraph and figure can be removed. The text is difficult to follow and it's unclear how the figure contributes to the discussion past the figures already displayed.

We agree and have removed Figure 13 and the associated discussion, thanks.

22) Lines 356–361: An alternative hypothesis is that a cold snap led to snowpack cooling. The more exposed site (Sap1) experienced greater cooling than the less exposed sites.

We have added the following sentence at the end of this paragraph: "Finally, as explained in section 4.3, the Sap1 site is more exposed to atmospheric conditions without the 'protective' effect of the canopy, making it more likely to respond to cold snaps for example."

23) Lines 365–366: Change "Based on our findings, CLASS successfully estimated snow density and CC. Although CLASS 365 reasonably predicted SWE, one cannot deny the fact that it underestimated observations" to "Based on our findings, CLASS successfully estimated snow density and CC, while underpredicting SWE."

We have modified the sentence accordingly, thanks.

24) Lines 367–371: Are these sentences suggesting class overestimates sublimation, leading to an underestimation of SWE? If so, please clarify.

The sentence is suggesting that CLASS is overestimating latent heat flux. This now reads as: "Based

on these recent studies, it seems fair to conclude that the overestimation of latent heat fluxes by CLASS could lead to SWE underestimation."

25) Lines 380–407: This section should be shortened significantly. I think it is sufficient to say the new snow density formulations led to persistent underprediction errors for the top layers, even when compaction is considered. This is likely a shortcoming of both the fresh density and compaction due to metamorphosis equations.

We believe this is an important section of this manuscript. We prefer to keep it as it is.

26) Lines 410–442: I really like the inclusion of the energy balance analysis, but these sentences contain novel results. Please keep the discussion parts here while moving the quantifications to an additional results subsection. (Also, please limit the use of "indeed" to start sentences.)

First, we would like to point out that the presence of the energy balance analysis is in response to a past request from the reviewers and that without them, this very relevant addition to the paper would have been impossible.

After careful rereading of this section, it appears that the vast majority of it contains analyses based on many figures from the results section and therefore, in our opinion, it is more appropriate to leave it in the discussion. There are certainly some small new results, but these are only intended to provide a very focused addition to the discussion. We therefore prefer to keep the section as it is now, hoping that the reviewer now shares our opinion.

Finally, you are right, there was a slightly exaggerated use of the word "indeed". We have either removed or replaced 3 of its 4 occurrences.

27) Line 417: Remove "at" from in front of "canopy"

Done, thanks.

28) Line 419: My take on this is that the 0.1 MJ m-2 difference is negligible given the observation precisions and model biases.

True, we removed the cross-reference here, thank.

29) Line 434: Please expand on this sentence. The canopy blocks incoming solar radiation (more CC) but also increases incoming longwave radiation (less CC) compared to an open site.

We have expanded this sentence, thanks.

30) Line 437: Please cite peer-reviewed literature on energy fluxes during rain-on-snow (e.g., Marks et al., 1998): https://doi.org/10.1002/(SICI)1099-1085(199808/09)12:10/11%3C1569::AID-HYP682%3E3.0.CO;2-L

We have cited this reference, thanks.

Reference

Alves, M., Nadeau, D. F., Music, B., Anctil, F. and Parajuli, A.: On the performance of the Canadian Land Surface Scheme driven by the ERA5 reanalysis over the Canadian boreal forest, J. Hydrometeorol., 21(6), 1383–1404, doi:10.1175/jhm-d-19-0172.1, 2020.

Seligman, Z. M., Harper, J. T. and Maneta, M. P.: Changes to snowpack energy state from spring storm events, Columbia River headwaters, Montana, J. Hydrometeorol., 15(1), 159–170, doi:10.1175/JHM-D-12-078.1, 2014.